# GeoRemover: Removing Objects and Their Causal Visual Artifacts

**Zixin Zhu**[1,2*] **Haoxiang Li**[2†] **Xuelu Feng**[1] **He Wu**[2] **Chunming Qiao**[1] **Junsong Yuan**[1]

[1]University at Buffalo  [2]Pixocial Technology

{zixinzhu, xuelufen, qiao, jsyuan}@buffalo.edu,
haoxiang.li@pixocial.com, heu199825@gmail.com

## Abstract

Towards intelligent image editing, object removal should eliminate both the target object and its causal visual artifacts, such as shadows and reflections. However, existing image appearance-based methods either follow strictly mask-aligned training and fail to remove these casual effects which are not explicitly masked, or adopt loosely mask-aligned strategies that lack controllability and may unintentionally over-erase other objects. We identify that these limitations stem from ignoring the causal relationship between an object's geometry presence and its visual effects. To address this limitation, we propose a geometry-aware two-stage framework that decouples object removal into (1) geometry removal and (2) appearance rendering. In the first stage, we remove the object directly from the geometry (e.g., depth) using strictly mask-aligned supervision, enabling structure-aware editing with strong geometric constraints. In the second stage, we render a photorealistic RGB image conditioned on the updated geometry, where causal visual effects are considered implicitly as a result of the modified 3D geometry. To guide learning in the geometry removal stage, we introduce a preference-driven objective based on positive and negative sample pairs, encouraging the model to remove objects as well as their causal visual artifacts while avoiding new structural insertions. Extensive experiments demonstrate that our method achieves state-of-the-art performance in removing both objects and their associated artifacts on two popular benchmarks. The project page is available at `https://buxiangzhiren.github.io/GeoRemover`.

## 1 Introduction

Object removal is a challenging computer vision task with applications in image editing and scene rendering, aiming to erase undesired objects as if they never present. Following the inpainting framework [1, 2, 3, 4], traditional strictly mask-aligned approaches [5, 6, 7, 8] assume that user specified mask fully covers the objects to be removed, thus only deal with the masked region while do not change the remained image. However, in real-world scenarios, objects often cast causal visual artifacts (e.g., shadows and reflections) onto surrounding regions, leading to illumination inconsistencies beyond the masked area. As illustrated in Fig. 1a, although the child is successfully removed, his shadow remains as the causal artifact. A simple solution to address such an associated artifact is to extend the object removal mask to cover these artifacts, but this places a significant burden on users, who must identify and annotate all subtle, detached, and ambiguous artifacts. As a result, this approach is neither scalable nor user-friendly.

Therefore, recent methods [9, 10, 11] assume a more practical and user-friendly setting where the input mask only covers the objects to remove, but the model implicitly infers and removes causal visual artifacts such as shadows and reflections in an intelligent way. For example, previous methods

---

*Work completed while the author was an intern at Pixocial Technology.
†Corresponding author.

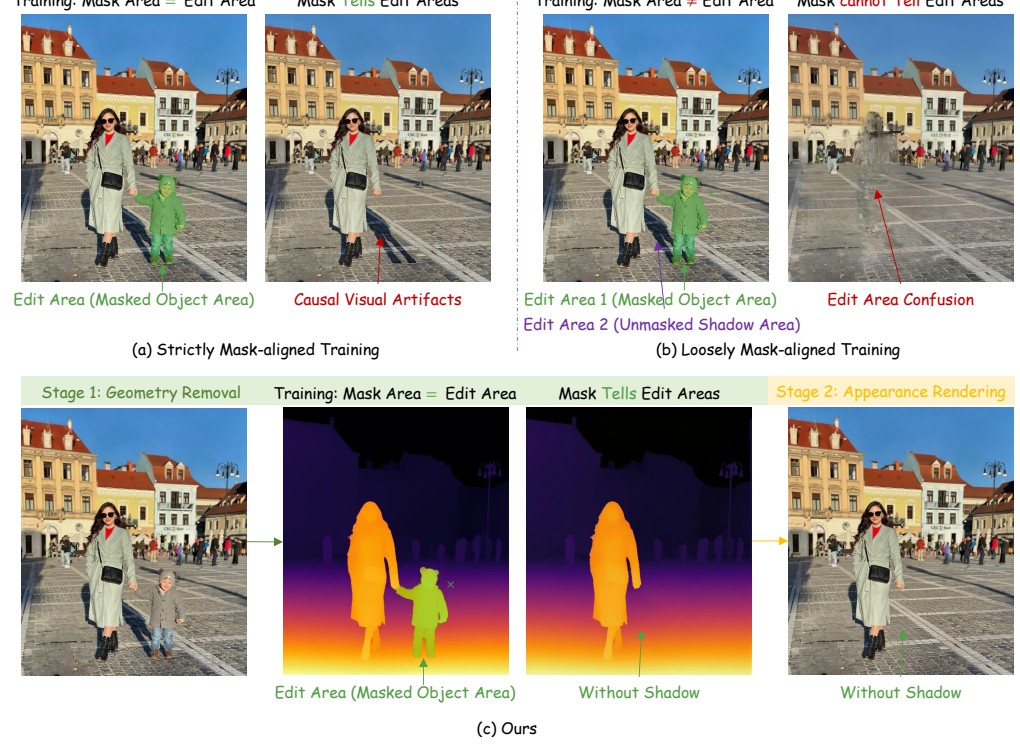

Figure 1: Comparison of object removal training paradigms. (a) Strictly mask-aligned training edits only masked regions but leaves causal visual artifacts (shadow) unaddressed. (b) Loosely mask-aligned training allows broader context-aware corrections but lacks clear guidance, leading to confusion and uncontrollable edits. (c) Our method decouples geometry and appearance for object removal: we first edit the scene geometric representation (in the form of a depth map) under strictly mask-aligned supervision, then render a realistic image where both objects and causal visual artifacts (shadow) are cleanly removed.

have attempted to adopt loosely mask-aligned training strategies, encouraging models to infer and correct inconsistencies beyond explicitly masked regions specified by the user. However, without a specific design to guide the editing, most of them heavily rely on paired training data, which hinders controllability. Compared to strictly mask-aligned training, where the mask explicitly defines which regions can be modified and which must be preserved, in loosely mask-aligned settings, both the masked and unmasked regions may require edits, but the model itself has no clear boundary guidance, leading to confusion about where modifications should occur. As shown in Fig. 1b, while the model successfully removes the child's shadow, it also mistakenly removes the nearby adult, resulting in unintended alterations to the scene.

The above challenges suggest that solely optimizing training strategies is insufficient to enable models to reason about causal visual effects. We make a key observation: these effects, such as shadows and reflections, are fundamentally caused by the object's geometry under specific lighting conditions. In other words, the *geometry presence is the cause*, and the *causal visual effects are its consequence*. Intuitively, if the object's presence is removed from the scene geometry, then its associated illumination effects should no longer exist.

This insight motivates us to rethink object removal as a causal reasoning process: we firstly modify the geometric representation (e.g., via modifying the depth maps) to remove the object presence from the scene geometry; then, we render a new image appearance based on the updated scene geometry, where causal visual artifacts will be naturally removed. This progressive design offers two key advantages. First, in the geometry removal stage, we can adopt strictly mask-aligned training: since causal visual artifacts do not need to be considered in the geometry domain, and object boundaries are clearly defined, thus the model can focus on removing only the object in masked region, making the task well-posed with strong supervision. This eliminates the risk of undesired modifications to unmasked regions. Second, in the rendering stage, the absence of the object naturally leads to the

removal of its associated artifacts. This implicitly enforces causal consistency by removing both the object and its visual artifacts. To enable this behavior, we train the rendering model with paired data: each pair consists of an image with the object and its causal effects (e.g., shadows or reflections), and the corresponding image shows the scene with both the object and its effects removed. By first localizing the removed object based on geometric differences, the rendering model can then establish the causal relationship between the object and its associated effects by analyzing the visual differences between the paired images. As shown in Fig. 1c, our method successfully removes both the object and its shadow, while preserving nearby unmasked content. Our contributions can be summarized as:

- We propose a new two-stage framework to leverage geometric representation to decouple object removal into geometry removal and appearance rendering. Based on our observation that the geometric representation is free from causal visual artifacts, our method erases masked objects from the scene geometry followed by the removal of their visual artifacts.

- To improve object removal quality in the geometric representation, we introduce a preference-guided loss to prevent the model from inserting unexpected structures.

- Compared to existing methods that utilize loosely mask-aligned training strategies to approach this problem in the same setting, which mostly suffer from loss of controllability and the unintended alteration issue, we demonstrate through experiments that the proposed framework improves the removal quality on two benchmark datasets.

## 2 Related work

**Object removal and inpainting.** Object removal is traditionally formulated as an inpainting problem, where the model fills a user-specified mask with realistic content [12, 13, 14, 15, 16, 17, 18]. Most approaches [19, 20, 7] trained in a strictly mask-aligned manner, enabling precise control over the masked region. For instance, ClipAway [5] leverages harmonized CLIP embeddings to guide removal, SmartEraser [21] and Inst-Inpaint [6] explore instruction-based or maskless generation. But they often leaving behind causal visual artifacts such as shadows and reflections. To address this, loosely mask-aligned methods [9, 10, 11] expand the removal scope beyond the user-defined mask, automatically cleaning surrounding artifacts. However, this comes at the cost of reduced controllability and increased risk of over-editing. Different from them, our method bridges the gap between strictly mask-aligned precision and loosely mask-aligned flexibility by decoupling geometry and appearance, allowing for structure-aware editing and implicit removal of causal visual artifacts.

**Geometry-aware generation.** Incorporating geometric priors such as depth maps has shown promise in editing [22], scene completion [23], and novel view synthesis [24, 25]. While many prior methods use geometry as auxiliary input during single-stage generation [26, 27], they typically entangle structure and appearance modeling. Unlike prior methods that use depth as auxiliary input, we decouple geometry and appearance into two stages: editing in depth space and rendering in RGB. This design enables controllable structure editing and implicit removal of causal visual artifacts.

**Diffusion preference optimization and human alignment.** Preference-guided training has emerged as a practical alternative to supervised learning for aligning generative models with human intent [28, 29, 30, 31]. For instance, DPO [28] extends preference optimization to diffusion models by learning from ranked pairs. Other works [29, 30, 31] explore human alignment through benchmark design, direct reward modeling, and multi-dimensional preference decomposition for text-to-image generation. Inspired by these ideas, we adopt a DPO-style strategy in our geometry removal stage: instead of human-annotated rankings, we define preferences based on geometric flow smoothness, encouraging plausible structure completion while suppressing spurious insertions.

## 3 Method

### 3.1 Problem formulation

Given an input image $I^- \in \mathbb{R}^{H \times W \times 3}$ and an object mask $M \in \{0, 1\}^{H \times W}$ indicating the region to be removed, our goal is to generate an output image $I^+ \in \mathbb{R}^{H \times W \times 3}$ in which the object has been cleanly erased, its contextual effects (e.g., shadows or reflections) are removed, and the background

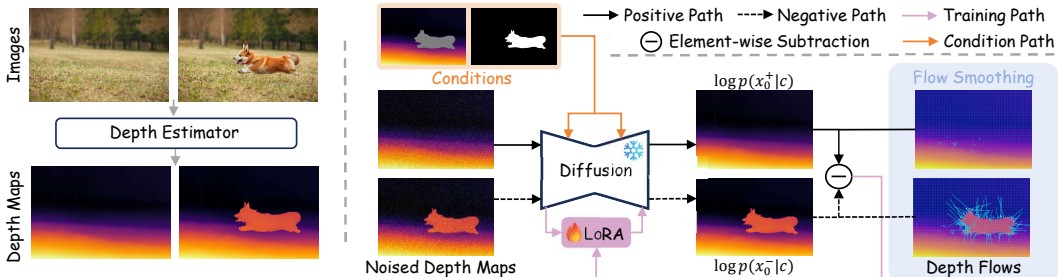

Figure 2: The training framework of Stage 1: Geometry Removal. Given an input image and object mask, we first estimate the geometric representation (in the form of a depth map) and construct a masked geometry input. The masked depth map, together with the mask, is then fed into a diffusion model to predict the edited geometry. To discourage structure insertion and encourage object removal, we construct two geometry completion paths: a positive path where the object is successfully removed with smooth depth flow, and a negative path where the object remains with sharp depth transitions. The model is trained to prefer the positive path and suppress the negative one.

is realistically restored. Most existing methods formulate this problem as a direct image-to-image transformation task, learning a mapping

$$I^+ = g(I^-, M), \tag{1}$$

where $g$ is a function that maps the masked image and mask to a completed result. However, as illustrated in Fig. 1a and Fig. 1b, such formulations often entangle geometric reasoning with appearance synthesis, making it difficult to control structural edits and leading to unintended modifications.

To address this, we approach this problem by decoupling it into two sub-tasks: (1) geometry removal, which modifies the geometric representation to eliminate the object while preserving the surrounding structure; and (2) appearance rendering, which synthesizes an RGB image consistent with the updated geometry from geometry removal. This decoupling allows us to separate structure-level editing from pixel-level synthesis, and enables causal visual artifacts to be implicitly corrected through geometry-aware rendering. Formally, we decompose the object removal process as

$$\underbrace{x_0^- = \mathcal{D}(I^-), \quad x_0^+ = s_\theta(x_0^-, M),}_{\text{Stage 1: geometry removal}} \quad \underbrace{I^+ = \mathcal{G}(I^-, x_0^-, x_0^+),}_{\text{Stage 2: appearance rendering}} \tag{2}$$

where $\mathcal{D}$ is a geometry estimator, $x_0^-$ is the estimated geometric representation of the input image, $x_0^+ = s_\theta(x_0^-, M)$ is the updated geometry predicted by the diffusion model $s_\theta$ under strictly mask-aligned supervision, and $\mathcal{G}$ synthesizes the final RGB output conditioned on the geometric transformation from $x_0^-$ to $x_0^+$ and the input image $I^-$.

### 3.2 Stage 1: geometry removal

**Geometry completion with strictly mask-aligned training.** In the first stage, our goal is to remove the target object by modifying the scene geometry, while preserving the surrounding structure. We use depth as the geometric representation in this work due to the efficiency and accuracy of recent depth estimation models. Geometry removal is performed in the depth domain, where causal visual artifacts such as shadows and reflections do not appear, making the task well-suited for strictly mask-aligned supervision. The overall training pipeline is illustrated in Fig. 2. Formally, given an input RGB image $I$, an estimated depth map $x_0$, and an object mask $M \in \{0, 1\}^{H \times W}$ indicating the removal region, the objective is to learn a model that predicts an edited depth map $\hat{x}_0$ where the object is removed within the masked region, while preserving geometry elsewhere. We enforce the constraint

$$\hat{x}_0(i, j) = x_0(i, j), \quad \forall(i, j) \text{ where } M(i, j) = 0. \tag{3}$$

A naive solution to this task is to treat the depth map as a colorized image and fine-tune a pre-trained diffusion-based image inpainting model for depth editing. To maximize the log-likelihood $\log p(x_0 \mid c)$, diffusion-based models optimize a denoising score matching objective, which minimizes the discrepancy between the model-predicted score $s_\theta(x_t, t, c)$ and the true score $\nabla_{x_t} \log p(x_t \mid x_0, c)$.

Here, $s_\theta$ is a parameterized score function, and $c = (M, (1 - M) * x_0)$ denotes the conditioning input (i.e., the object mask and the masked depth map). The score matching loss is

$$\mathcal{L}_{\text{DSM}}(x_0, c) = \mathbb{E}_{t,\epsilon} \left[ w(t) \left\| s_\theta(x_t, t, c) - \nabla_{x_t} \log p(x_t \mid x_0, c) \right\|^2 \right], \tag{4}$$

where $x_t$ is a noisy sample generated from $x_0$ and $w(t)$ is a weighting function over timesteps.

**Preference-guided geometry completion via DPO.** However, when applying this baseline directly, we observe that the model often hallucinates new structures within the masked region, as shown in the second row of Fig. 3a. Rather than recovering a coherent surface, it tends to insert unrealistic geometry that does not align with the surrounding structure. We hypothesize that this behavior arises from the lack of geometry-aware constraints: without explicit structural supervision, the model cannot distinguish between completing missing surfaces and generating new, implausible content. To avoid hallucinating new content within the object removal region, inspired by recent advances in Direct Preference Optimization (DPO) [28], we propose to model geometry removal through a reward-based framework. DPO aims to align model outputs with user preferences by optimizing over ranked sample pairs, rather than relying solely on explicit ground-truth labels.

In our setting, we adopt a similar philosophy: we define preferences over geometry, where the depth in the masked region that does not contain the object is what we prefer, and the depth that includes the object is what we do not prefer. Ideally, a preferred depth map should be locally smooth inside the mask, with minimal abrupt depth changes that would otherwise indicate the presence of an object. As shown in Fig. 2, when the masked region contains an object (e.g., a dog), the depth flow, defined as the spatial gradient of depth values, exhibits sharp discontinuities due to the object's geometry. In contrast, when the object is successfully removed from the mask, the depth flow approaches zero, indicating a smooth and coherent surface. Therefore, we consider low depth flow within the mask as a key signal for realistic and desirable geometry.

We define the reward as a monotonic function of $\log p(x_0 \mid c)$, based on flow difference between the predicted depth map $\hat{x}_0$ and ground-truth $x_0$, measured by the flow loss $\mathcal{L}_{\text{flow}}(\hat{x}_0, x_0)$. The reward is

$$r(c, x_0) = -\mathcal{L}_{\text{flow}}(\hat{x}_0, x_0) = f \left( \log p(x_0 \mid c) \right), \quad \text{with } f' > 0. \tag{5}$$

Then we introduce how we define the flow loss $\mathcal{L}_{\text{flow}}(\hat{x}_0, x_0)$. Specifically, let $d_{ij}$ denote the depth value at pixel $(i, j)$. The flow at pixel $(i, j)$ is defined as the first-order spatial gradient

$$F_{ij}(x) = \left\{ |d_{i+1,j} - d_{i,j}|, \ |d_{i,j+1} - d_{i,j}| \right\}, \tag{6}$$

which captures local depth transitions in horizontal and vertical directions. Then the flow loss is defined as the average of per-pixel absolute differences between the predicted flow and the ground-truth flow. It is

$$\mathcal{L}_{\text{flow}}(\hat{x}_0, x_0) = \frac{1}{|\Omega|} \sum_{(i,j) \in \Omega} \left\| F_{ij}(\hat{x}_0) - F_{ij}(x_0) \right\|_1, \tag{7}$$

where $\Omega$ denotes the set of valid pixels. With the reward $r(c, x_0)$, to model preference between completions, we assume access to ranked sample pairs $(x_0^+, x_0^-)$, indicating that $x_0^+$ is preferred over $x_0^-$ under the same conditioning $c$. As illustrated in Fig. 2, we refer to these as the positive and negative geometry paths, respectively. We adopt the Bradley-Terry (BT) model to express the preference probability

$$\mathcal{L}_{\text{BT}} = -\mathbb{E}_{c,x_0^+,x_0^-} \left[ \log \sigma \left( r(c, x_0^+) - r(c, x_0^-) \right) \right]. \tag{8}$$

The final loss $\mathcal{L}$ combines the standard diffusion loss with the preference-guided objective, which is

$$\mathcal{L} = \mathcal{L}_{\text{DSM}} + \lambda \, \mathcal{L}_{\text{BT}}, \tag{9}$$

where $\lambda$ balances score-based denoising supervision and geometry-consistent preference learning.

### 3.3 Stage 2: appearance rendering

In the second stage, our goal is to generate a realistic RGB image that reflects the scene after object removal, as defined by the updated geometry from Sec. 3.2. We formulate this task as a conditional image translation problem, where the appearance of the output image is controlled by the geometric

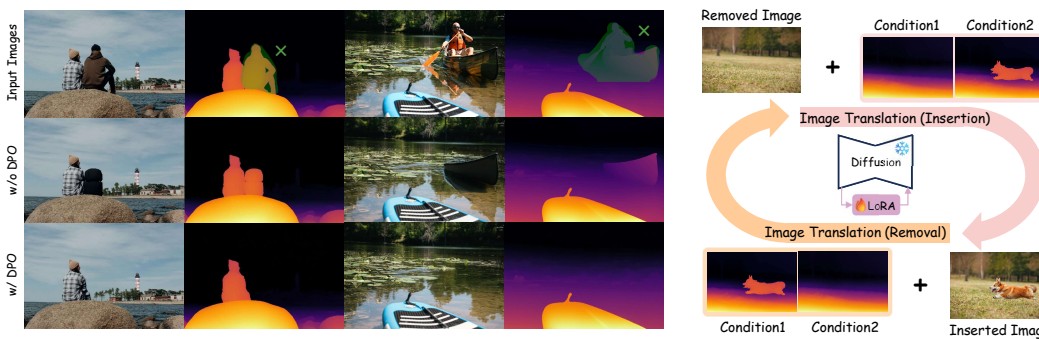

(a) Effect of direct preference optimization (DPO) in Stage 1.   (b) Stage 2: Appearance rendering.

Figure 3: (a) We compare model outputs trained with and without our DPO objective (i.e., $\mathcal{L}_{\text{BT}}$). Without DPO, the model often inserts or retains undesired content in the masked region, leading to unrealistic geometry. With DPO, the model learns to prefer geometry completions that successfully remove the object while preserving surrounding structures. (b) The training framework of Stage 2. Given geometry-aware conditions (e.g., depth maps), we train a diffusion model to perform image translation for both object removal and insertion.

transformation between two depth maps. As illustrated in Fig. 3b, the model takes as input a masked RGB image and two geometry-aware conditions: Condition1 and Condition2. Both conditions are represented as depth maps: Condition1 encodes the geometry of the input image (e.g., with an object present), while Condition2 defines the target geometry (e.g., with the object removed). The difference between Condition1 and Condition2 specifies the structural transformation to be performed. For example, if Condition2 removes an object that exists in Condition1, the model is guided to erase that object and inpaint a realistic background. Conversely, if Condition2 introduces a localized depth discontinuity compared to a flat Condition1, the model learns to insert a visually plausible object.

Formally, we define $\mathcal{G}$ as a geometry-conditioned image translation model based on a diffusion backbone. The model takes as input a RGB image and a pair of depth maps indicating the geometry before and after editing. Let $I^-$ and $I^+$ denote the input and output images, and let $x_0^-$ and $x_0^+$ represent the corresponding depth maps before and after editing. The model learns to perform both object removal and insertion through the following bidirectional formulation:

$$I^+ = \mathcal{G}(I^- \mid x_0^-, x_0^+), \quad I^- = \mathcal{G}(I^+ \mid x_0^+, x_0^-). \tag{10}$$

Further details of the loss formulation for this stage are provided in Appendix A.

## 4   Experiments

**Implementation details.**   The depth estimator $\mathcal{D}$ is implemented using Depth Anything [32]. For both geometry removal model ($s_\theta$) and appearance rendering model ($\mathcal{G}$), we adopt FLUX.1-Fill-dev [8] as the pre-trained diffusion backbone, and apply LoRA [33] fine-tuning with a rank of 64. All images are processed at a resolution of 1024×1024. For both stages, we use a batch size of 24, a learning rate of $1 \times 10^{-4}$, and a guidance scale of 1.0. The text prompt "a beautiful scene" is used during both training and inference. Stage 1 is trained for 17,000 steps on 8 NVIDIA H100 GPUs, taking approximately 24 hours, while Stage 2 requires around 60 hours for the same number of steps.

**Datasets & Metrics.**   We use the training set from the RORD [34] dataset as our primary training data. RORD is a large-scale real-world object removal dataset consisting of 516,705 images captured under 3,447 unique indoor scenes. Each scene contains paired images with and without the target object, along with manually annotated object masks. The dataset is designed to support training and evaluation for object removal and scene completion tasks in realistic environments. For evaluation, we follow prior works such as SmartEraser [21] and OmniEraser [9], We use both RORD-Val and RemovalBench [9] as our primary benchmarks. Moreover, we follow prior works [35, 9] and adopt a set of metrics to evaluate image generation quality. We use Frechet Inception Distance (FID)[36], CLIP Maximum Mean Discrepancy (CMMD)[37], Aesthetic Score (AS)[38], Learned Perceptual Image Patch Similarity (LPIPS)[39] and Peak Signal-to-Noise Ratio (PSNR) [40].

Table 1: Comparison with state-of-the-art methods on RemovalBench and RORD-Val.

| Method | RemovalBench | | | | | RORD-Val | | | | |
|---|---|---|---|---|---|---|---|---|---|---|
| | FID ↓ | CMMD ↓ | LPIPS ↓ | PSNR ↑ | AS ↑ | FID ↓ | CMMD ↓ | LPIPS ↓ | PSNR ↑ | AS ↑ |
| ZITS++ [41] | 108.38 | 0.374 | 0.158 | 19.62 | 4.56 | 107.44 | 0.448 | 0.274 | 21.17 | 4.12 |
| MAT [19] | 123.78 | 0.366 | 0.164 | 17.88 | 4.51 | 136.53 | 0.455 | 0.281 | 19.18 | 4.38 |
| LaMa [42] | 99.88 | 0.351 | 0.156 | 18.72 | 4.55 | 100.21 | 0.294 | 0.229 | 20.50 | 4.23 |
| RePaint [20] | 102.65 | 0.741 | 0.378 | 19.86 | 4.38 | 114.64 | 2.345 | 0.525 | 17.68 | 4.71 |
| BLD [43] | 128.66 | 0.553 | 0.233 | 17.43 | 4.39 | 224.61 | 0.862 | 0.273 | 17.13 | 4.74 |
| LDM [7] | 108.79 | 0.365 | 0.157 | 19.24 | 4.47 | 128.19 | 0.506 | 0.221 | 19.02 | 4.12 |
| SD-Inpaint [7] | 119.60 | 0.419 | 0.274 | 17.02 | 4.48 | 143.69 | 0.494 | 0.308 | 16.83 | 4.61 |
| SDXL-Inpaint [7] | 104.97 | 0.398 | 0.187 | 17.87 | 4.63 | 147.01 | 0.460 | 0.210 | 17.69 | 4.76 |
| BrushNet [35] | 120.97 | 0.549 | 0.191 | 18.68 | 4.63 | 234.87 | 0.745 | 0.293 | 16.51 | 4.41 |
| FLUX.1-Fill [8] | 115.79 | 0.487 | 0.193 | 17.12 | 4.59 | 141.39 | 0.450 | 0.217 | 18.50 | 4.55 |
| PowerPaint [44] | 114.55 | 0.392 | 0.240 | 18.25 | 4.56 | 102.33 | 0.408 | 0.241 | 18.29 | 4.38 |
| CLIPAway [5] | 108.40 | 0.272 | 0.254 | 18.78 | 4.48 | 81.28 | 0.545 | 0.278 | 16.36 | 4.19 |
| Attentive-Eraser [45] | 55.49 | 0.232 | 0.146 | 20.60 | 4.50 | 96.77 | 0.233 | 0.221 | 20.24 | 4.77 |
| OmniEraser [9] | 39.52 | 0.208 | 0.133 | 21.11 | **4.66** | 43.71 | **0.153** | 0.166 | 22.13 | **4.99** |
| Ours | **29.88** | **0.089** | **0.124** | **25.52** | 4.54 | **31.15** | 0.182 | **0.103** | **23.70** | 4.69 |

Table 2: Ablation study on RORD-Val to evaluate the effectiveness of our design components. "Insert." denotes the percentage of cases where a new object is wrongly inserted into the removal region.

| Method | FID ↓ | CMMD ↓ | LPIPS ↓ | PSNR ↑ | AS ↑ | Insert. ↓ |
|---|---|---|---|---|---|---|
| One-Stage | 56.24 | 0.577 | 0.315 | 17.52 | 4.27 | 2.81% |
| Two-Stage w/o DPO | 34.24 | 0.230 | 0.131 | 22.81 | 4.51 | 5.09% |
| Two-Stage w/ DPO | **31.15** | **0.182** | **0.103** | **23.70** | **4.69** | **1.48%** |

Table 3: Geometry removal accuracy (MAE in masked region) on RORD-Val.

| Method | MAE ↓ |
|---|---|
| Input depth | 0.0827 |
| Two-Stage w/o DPO | 0.0490 |
| Two-Stage w/ DPO | **0.0387** |

## 4.1 Comparison with SOTA methods

We compare our method against state-of-the-art approaches on the RemovalBench and RORD-Val datasets, as shown in Tab. 1. These baselines fall into two categories: **strictly mask-aligned methods** and **loosely mask-aligned methods**. Strictly mask-aligned methods [20, 7, 19, 5, 45] are limited in their ability to handle contextual effects, since they are confined to the object region defined by the user. In contrast, loosely mask-aligned methods can adaptively clean surrounding regions affected by the object. Among loosely mask-aligned approaches, OmniEraser [9] is the only open-source method that supports causal visual artifacts removal. Although models like OmniPaint [11] and ObjectDrop [10] also aim to remove such effects, their code and models are not publicly available, and we were therefore unable to include them in our evaluation. Across both benchmarks, our method consistently achieves the best scores in FID, CMMD, LPIPS, and PSNR, demonstrating superior visual quality and structure preservation in the removed regions.

## 4.2 Ablation study and discussion

**Is the two-stage design necessary?**    Compared to prior one-stage approaches, our method introduces two key innovations: (1) a two-stage design that explicitly decouples geometry and appearance, and (2) the incorporation of geometric cues such as depth to guide object removal. To fairly isolate the contribution of the two-stage architecture, we construct a one-stage version of our method that also takes the masked RGB image and masked depth map as input to the diffusion model. This ensures that both models operate on the same input modalities, and any performance gap can be attributed to the architectural difference. As shown in Fig. 4, despite access to depth information, the one-stage model often produces ambiguous or distorted edits due to the lack of explicit geometric guidance. Quantitative results in Tab. 2 further confirm that the one-stage variant consistently underperforms the two-stage models across multiple metrics. This supports the claim that it is the decoupled design, rather than merely the inclusion of depth, that enables our model to reason more effectively. While the

Table 4: Removal performance of causal artifacts on CausRem.

| Method | IoU% ↑ |
|---|---|
| OmniEraser [9] | 68.29 |
| Ours | **73.76** |

Table 5: Ablation study on the RORD-Val dataset comparing unidirectional and bidirectional rendering strategies in Stage 2.

| Method | FID ↓ | CMMD ↓ | LPIPS ↓ | PSNR ↑ | AS ↑ |
|---|---|---|---|---|---|
| Unidirectional rendering | 38.43 | 0.215 | 0.136 | 23.58 | 4.19 |
| Bidirectional rendering | **31.15** | **0.182** | **0.103** | **23.70** | **4.69** |

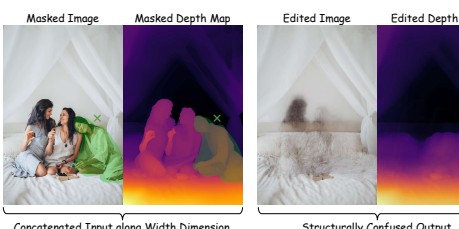

(a) Results from our one-stage model.

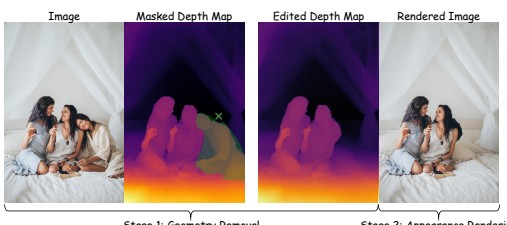

(b) Results from our two-stage model

Figure 4: Comparison between our one-stage and two-stage object removal strategies. Two-stage design improves edit quality by separating geometry reasoning from appearance generation.

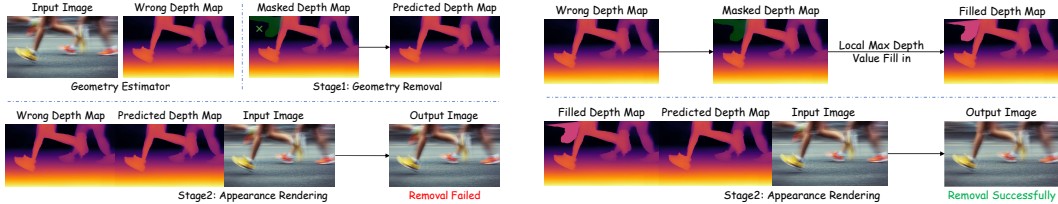

(a) Failure cases under motion blur conditions.    (b) Improved results after applying Fill-in strategy.

Figure 5: Depth errors caused by motion blur result in removal failure. Applying a simple *Fill-in* strategy within the mask restores geometric contrast and yields correct removal.

two-stage design increases runtime, the gains in controllability and illumination consistency justify the cost.

**How does the DPO strategy help our model achieve better removal?**    To evaluate the effect of Direct Preference Optimization (DPO), we compare our two-stage model with and without DPO supervision ($\mathcal{L}_{\text{BT}}$). As shown in Tab. 2, DPO significantly reduces the "Insert." rate from 5.08% to 1.48%, indicating its effectiveness in suppressing semantic hallucinations. This result suggests that preference-driven learning helps the model better align with human expectations for clean and plausible object removal.

**How accurate is geometry removal in Stage 1?**    To evaluate geometry removal quality, we compute the mean absolute error (MAE) between the predicted and ground-truth depth maps within the masked region, which corresponds to the object intended for removal. For reference, we also report the MAE between the input depth map and ground-truth depth map in the same region, providing a baseline for understanding the original geometric discrepancy. As shown in Tab. 3, our two-stage model significantly reduces the depth error within the masked region, and incorporating DPO further improves removal precision.

**How effectively does our model remove causal visual artifacts?**    To evaluate our method's ability to remove causal visual artifacts, we construct the **CausRem** benchmark, consisting of 200 real-world images (100 with shadows, 100 with reflections). Each image is manually annotated with object masks and corresponding artifact masks (shadows or reflections). Representative samples are provided in Appendix E. To estimate where the model implicitly removes such artifacts, we compute the pixel-wise absolute difference between the input and output within the annotated artifact regions. A threshold is then applied to identify significantly altered pixels, indicating predicted residue areas.

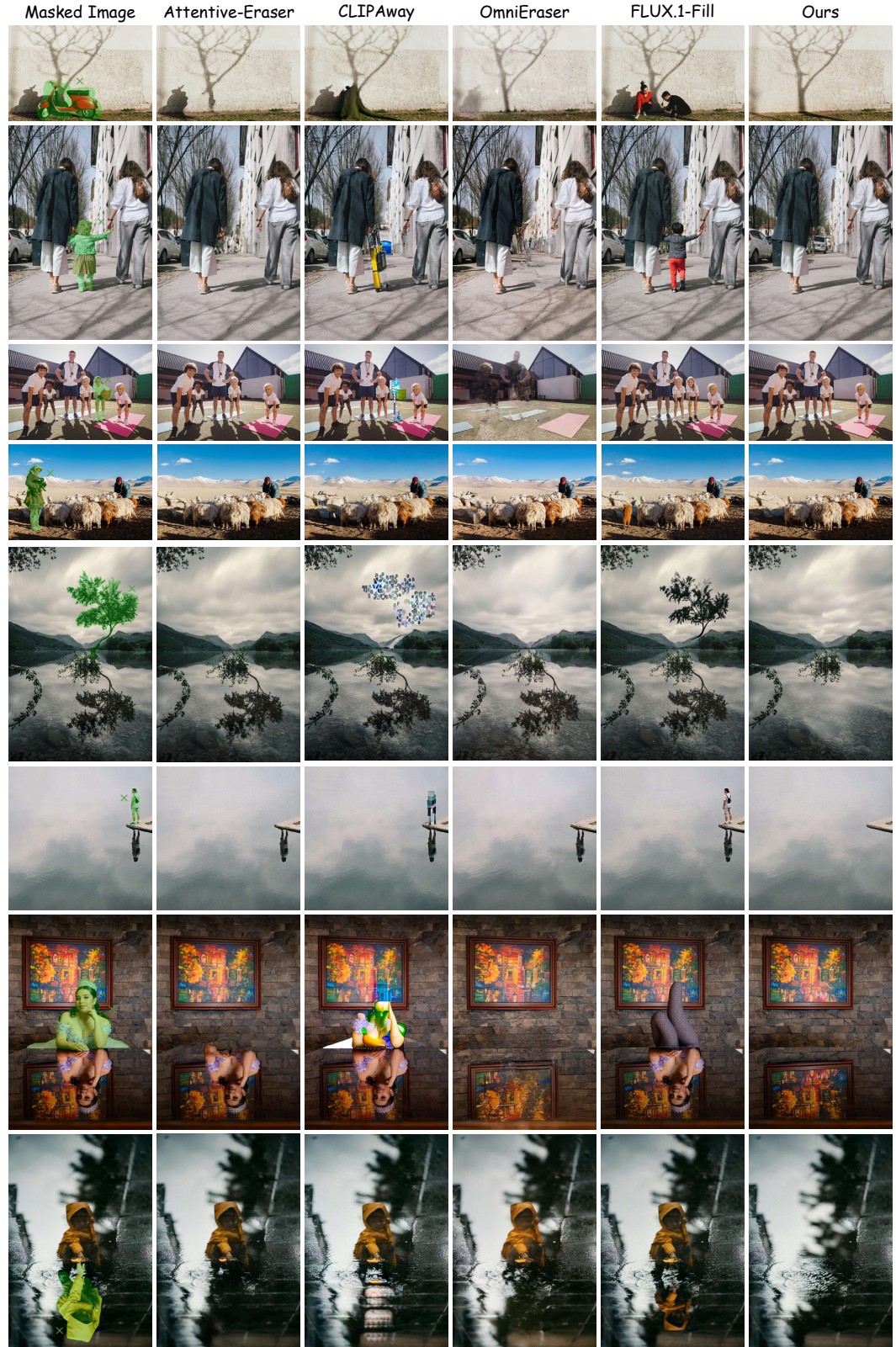

Figure 6: Qualitative comparison with state-of-the-art methods on CausRem.

To set a robust threshold, we analyze the boundary regions of the annotated causal visual artifact masks, where pixel values transition between the artifact and the background. We find the average difference in these regions to be approximately 20, which we adopt as a fixed global threshold. We evaluate the predicted residue regions using IoU against the ground truth. As shown in Tab. 4, our method achieves 73.76% IoU, outperforming the previous state-of-the-art OmniEraser [9] at 68.29%.

**Does bidirectional rendering improve performance in Stage 2?**  Tab. 5 demonstrates that bidirectional rendering enhances Stage 2 performance by promoting more precise alignment between the refined geometry and the synthesized image. By enforcing consistency in both removal and insertion rendering directions, the model is encouraged to maintain structural coherence throughout the image, leading to improved visual quality and reduced artifacts in the final output.

### 4.3  Qualitative comparison on CausRem

Fig. 6 shows a visual comparison of object and causal artifact removal (e.g., shadows, reflections) on the CausRem dataset. Compared with state-of-the-art methods, our approach yields cleaner results. Prior methods like Attentive-Eraser and CLIPAway often leave shadows or blur the background, while OmniEraser may distort nearby textures. In contrast, our method removes both objects and their effects cleanly by leveraging geometry-guided rendering, preserving background structure.

## 5  Failure case

Despite the overall robustness of our two-stage design, failures can still arise when the geometric signal inside the mask is weak or unreliable. Typical situations include fast motion, translucency, specular/reflective surfaces, occlusions, or low texture, where the "geometry-removed" depth becomes nearly indistinguishable from the input (Fig. 5a). Because Stage 2 identifies removal targets by differencing these two depth maps, the lack of geometric contrast prevents it from triggering removal.

We address this with a simple *Local Max Depth Fill-in*: for masked pixels lacking reliable estimates, we propagate the maximum depth from a small local neighborhood (e.g., $10\times10$ pixels). This lightweight completion restores boundary contrast and enables Stage 2 to remove the target while preserving a coherent background (Fig. 5b). Moreover, we provide additional challenging cases in Appendix D, including transparent and reflective surfaces, self-emitting scenes, and failures from incomplete masks, together with practical mitigations.

## 6  Conclusion

We present a geometry-aware, two-stage framework for object removal that effectively handles both the primary object and its associated causal visual artifacts. By decoupling the task into geometry removal and appearance rendering, our method achieves precise structural editing and seamless image restoration. Extensive experiments across multiple benchmarks validate that our approach outperforms existing methods in both quantitative metrics and visual quality, especially in challenging cases involving shadows and reflections.

**Broader impacts.**  Our work advances the controllability and accuracy of object removal systems, which can benefit applications in autonomous driving, AR/VR content editing, and photo restoration. However, the enhanced controllability of visual content manipulation also raises potential risks such as deepfakes. Mitigating these risks requires responsible deployment, provenance tracking, and clear guidelines for model usage.

## Acknowledgments

This work is supported in part by the National Science Foundation (NSF) and the Institute of Education Sciences (IES) through Award #2229873 (AI Institute for Transforming Education for Children with Speech and Language Processing Challenges) and NSF CNS 2413876 and CNS-2120369.

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

# NeurIPS Paper Checklist

The checklist is designed to encourage best practices for responsible machine learning research, addressing issues of reproducibility, transparency, research ethics, and societal impact. Do not remove the checklist: **The papers not including the checklist will be desk rejected.** The checklist should follow the references and follow the (optional) supplemental material. The checklist does NOT count towards the page limit.

Please read the checklist guidelines carefully for information on how to answer these questions. For each question in the checklist:

- You should answer [Yes] , [No] , or [NA] .
- [NA] means either that the question is Not Applicable for that particular paper or the relevant information is Not Available.
- Please provide a short (1–2 sentence) justification right after your answer (even for NA).

**The checklist answers are an integral part of your paper submission.** They are visible to the reviewers, area chairs, senior area chairs, and ethics reviewers. You will be asked to also include it (after eventual revisions) with the final version of your paper, and its final version will be published with the paper.

The reviewers of your paper will be asked to use the checklist as one of the factors in their evaluation. While "[Yes] " is generally preferable to "[No] ", it is perfectly acceptable to answer "[No] " provided a proper justification is given (e.g., "error bars are not reported because it would be too computationally expensive" or "we were unable to find the license for the dataset we used"). In general, answering "[No] " or "[NA] " is not grounds for rejection. While the questions are phrased in a binary way, we acknowledge that the true answer is often more nuanced, so please just use your best judgment and write a justification to elaborate. All supporting evidence can appear either in the main paper or the supplemental material, provided in appendix. If you answer [Yes] to a question, in the justification please point to the section(s) where related material for the question can be found.

IMPORTANT, please:

- **Delete this instruction block, but keep the section heading "NeurIPS Paper Checklist",**
- **Keep the checklist subsection headings, questions/answers and guidelines below.**
- **Do not modify the questions and only use the provided macros for your answers**.


## A Additional details about stage 2: appearance rendering

As described in Sec. 3.3, the appearance rendering model $\mathcal{G}$ learns to translate an input image containing an object into a realistic RGB output where both the object and its associated causal visual artifacts (e.g., shadows and reflections) are removed. This translation is guided by geometric transformations predicted in Stage 1. Rather than using separate conditioning vectors, we train $\mathcal{G}$ as a direct image-to-image diffusion model by concatenating the input image and geometry maps into a single tensor. To enable this, we directly use the colorized depth maps $x_0^+$ and $x_0^-$ produced in Stage 1, which are already represented as 3-channel RGB-like images. This allows us to concatenate them directly with the RGB image $I^-$ along the width dimension, forming a single composite input to the diffusion model.

The input to the appearance rendering model $\mathcal{G}$ is constructed by concatenating a masked RGB image with its geometric representations before and after editing. Specifically, we define two composite inputs for bidirectional training.

For object removal, the input is:

$$I^{\text{removal}} = \text{Concat}(I^-, x_0^+, x_0^-), \quad I^{\text{removal}} \in \mathbb{R}^{H \times (3W) \times 3}, \tag{11}$$

and the corresponding target is:

$$I^{\text{insert}} = \text{Concat}(I^+, x_0^+, x_0^-), \quad I^{\text{insert}} \in \mathbb{R}^{H \times (3W) \times 3}. \tag{12}$$

For the insertion direction, the roles of input and target are reversed, i.e., $(I^{\text{insert}}, I^{\text{removal}})$ forms the training pair. This bidirectional setup allows the same model to learn both removal and insertion through a unified diffusion process. The geometry maps $x_0^+$ and $x_0^-$ are placed on the left side of the RGB image, enabling $\mathcal{G}$ to directly observe spatial geometric changes. Since both depth maps are colorized 3-channel tensors, they can be processed jointly with the RGB image without additional modality-specific encoders. The model $\mathcal{G}$ is trained using a standard denoising score matching loss

$$
\begin{aligned}
\mathcal{L}_{\text{render}} = \mathbb{E}_{t,\epsilon} \left[ w(t) \left\| \mathcal{G}(I_t^{\text{removal}}, t) - \nabla_{I_t^{\text{removal}}} \log p(I_t^{\text{removal}} \mid I^{\text{insert}}) \right\|^2 \right] \\
+ \mathbb{E}_{t,\epsilon} \left[ w(t) \left\| \mathcal{G}(I_t^{\text{insert}}, t) - \nabla_{I_t^{\text{insert}}} \log p(I_t^{\text{insert}} \mid I^{\text{removal}}) \right\|^2 \right].
\end{aligned}
\tag{13}
$$

where $I_t$ is a noisy version of $I$ at diffusion timestep $t$, and $w(t)$ is a predefined weighting function. By jointly training on both directions, $\mathcal{G}$ learns to perform appearance synthesis conditioned on structured geometry edits, ensuring that generated results align with both scene content and layout.

## B Additional qualitative comparison

To further validate the effectiveness of our geometry-aware framework, we present additional qualitative results on the CausRem dataset, which contains real-world scenes involving shadows and reflections caused by removed objects.

As shown in Fig. 7, our method successfully removes both the object and its associated shadow, while other methods either retain shadow residues or introduce undesired distortions in unmasked regions. This highlights our method's ability to preserve unmasked content while achieving consistent object removal.

In Fig. 8, we provide additional comparisons in scenes with reflective surfaces. While baseline methods often fail to fully remove reflections or generate artifacts, our model leverages geometry-guided rendering to produce coherent appearances without explicit reflection modeling.

These results support our key insight: by removing the object structure in the geometric domain and rendering appearances based on updated geometry, our framework can implicitly eliminate causal visual artifacts and maintain visual consistency in challenging real-world scenarios.

## C Additional perceptual metrics

In Tab. 6, we additionally report SSIM [46], DISTS [47], DreamSim [48], FLIP [49], and CLIP-IQA [50] on RemovalBench and RORD-Val.

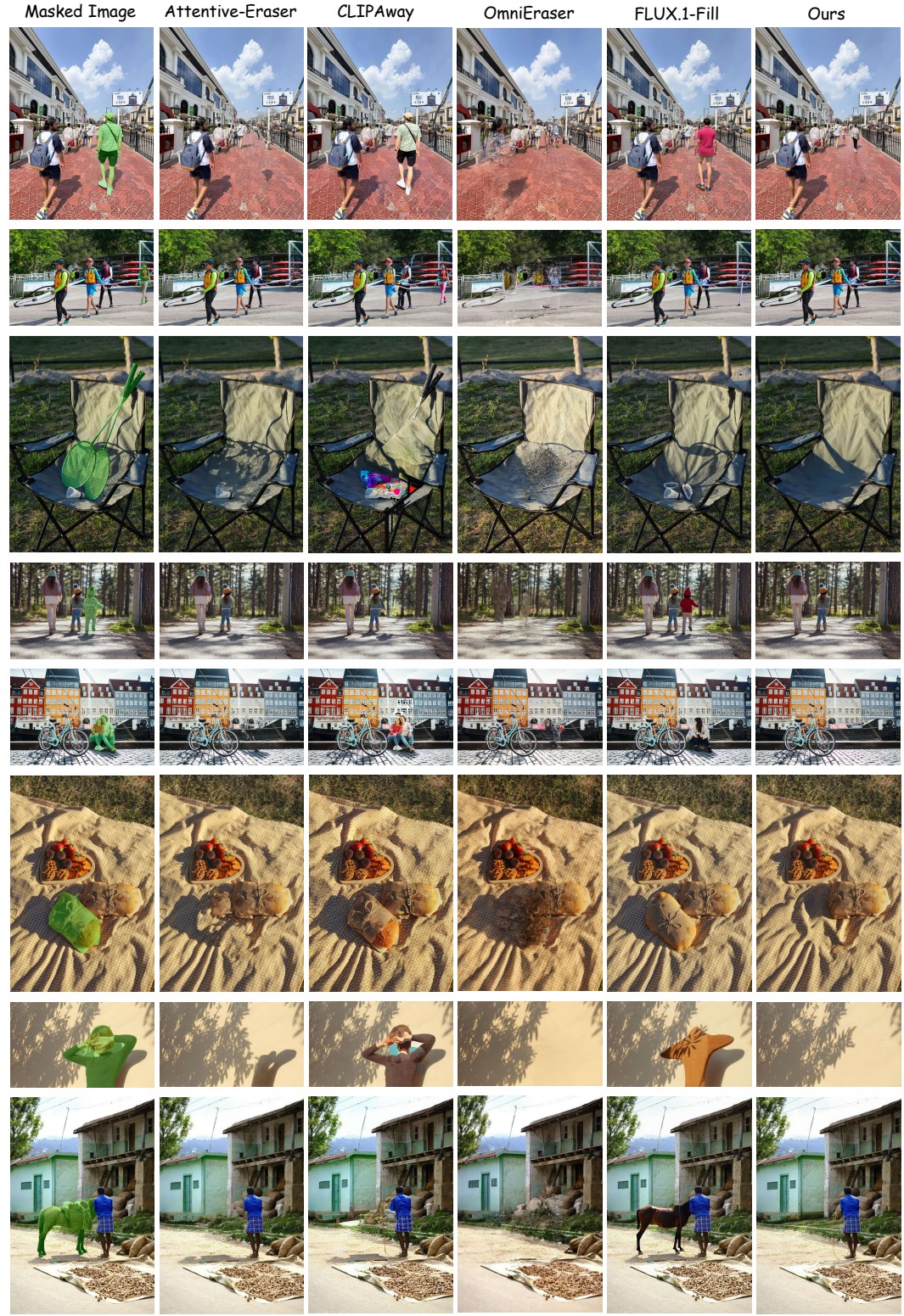

Figure 7: Qualitative comparison on CausRem highlighting shadow removal performance.

| Masked Image | Attentive-Eraser | CLIPAway | OmniEraser | FLUX.1-Fill | Ours |
|---|---|---|---|---|---|

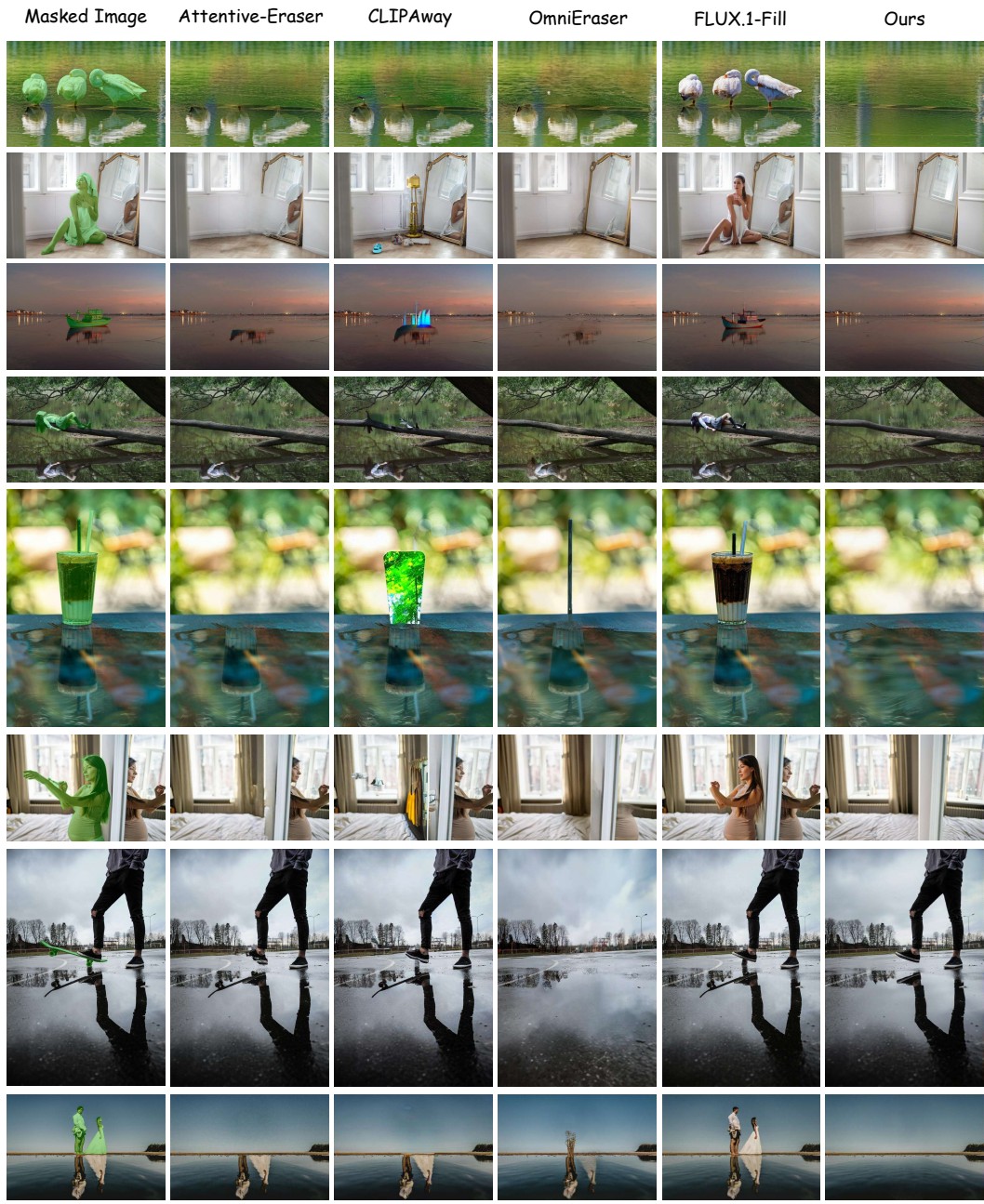

Figure 8: Qualitative comparison on CausRem highlighting reflection removal performance.

## D Additional failure and challenging cases

**Stage 2: transparent/reflective and self-emitting scenes.** The third row in Fig. 9a shows semi-transparent objects on a reflective table. After removing one cup, the direct reflection is removed as well, while residual color bleeding remains in nearby reflections influenced by the removed cup. As shown in the second row of Fig. 9a, for self-emitting objects (e.g., colored bulbs), Stage 2 may hallucinate a diffuse glow rather than cleanly removing the illumination. This behavior is undesirable for removal, yet it indicates that the renderer has learned meaningful correlations between light and its source, which could be useful if properly constrained.

Table 6: Perceptual metrics on **RemovalBench** and **RORD-Val**. ↑ higher is better, ↓ lower is better.

| Method | RemovalBench | | | | | RORD-Val | | | | |
|---|---|---|---|---|---|---|---|---|---|---|
| | SSIM↑ | DISTS↓ | DreamSim↓ | FLIP↓ | CLIP-IQA↑ | SSIM↑ | DISTS↓ | DreamSim↓ | FLIP↓ | CLIP-IQA↑ |
| CLIPAway [5] | 0.6298 | 0.1656 | 0.1572 | 0.1175 | **0.4973** | 0.6074 | 0.1580 | 0.1304 | 0.1645 | **0.7986** |
| Attentive-Eraser [45] | 0.7084 | 0.1168 | 0.0536 | **0.0854** | 0.4790 | 0.7186 | 0.1243 | 0.0878 | 0.1174 | 0.7270 |
| OmniEraser [9] | 0.6367 | 0.1277 | 0.0539 | 0.1084 | 0.4339 | 0.6071 | 0.1325 | 0.0675 | 0.1524 | 0.6646 |
| **Ours** | **0.7367** | **0.0770** | **0.0304** | 0.0863 | 0.4146 | **0.8248** | **0.0798** | **0.0459** | **0.1026** | 0.7807 |

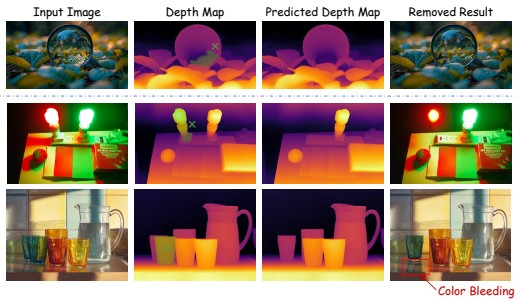

(a) Challenging scenes for Stage 2: transparent or semi-transparent objects and self-emitting (light-source) cases.

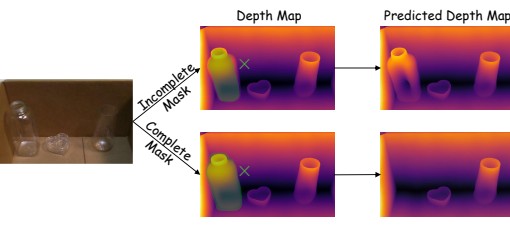

(b) Stage 1 failure under incomplete mask: complete versus partial masks yield success versus hallucinated completion.

Figure 9: Challenging cases and failure analysis. (a) Residual color bleeding and hallucinated glow can appear in reflective or lighting scenes. (b) Incomplete masks confuse Stage 1; simple dilation or stronger segmentation mitigates this.

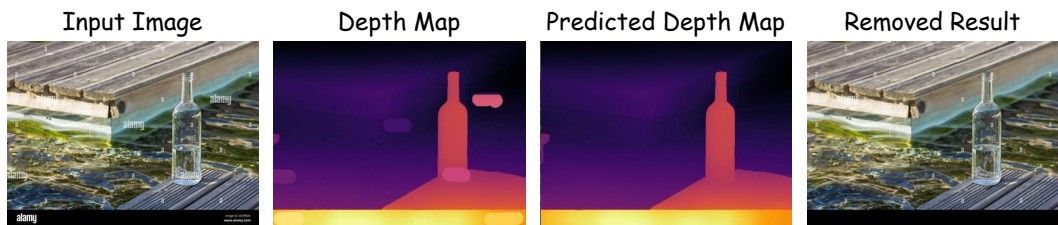

Figure 10: Watermark removal with a pseudo–depth cue. Applying Local Max Depth Fill-in within the watermark mask creates sufficient geometric contrast for Stage 2 to detect and remove the watermark across both the lake and dock.

**Stage 1: incomplete masks.** Figure 9b contrasts complete and partial masks for a semi-transparent bottle. With a complete mask, Stage 1 removes the geometry as expected. With a partial mask, the model attempts to complete the object, producing a hallucinated extra bottle. In practice this can be avoided by simple mask dilation or by using stronger segmentation models (e.g., SAM2) to provide complete masks.

**Watermark removal.** Beyond geometry-related artifacts, our framework can handle scene-wide watermarks with a light modification (Fig. 10). In this example, the watermark spans both the lake surface and wooden planks but lacks a reliable depth estimate. We apply the same *Local Max Depth Fill-in* inside the watermark mask—assigning each masked pixel the maximum depth from a small local neighborhood—as a pseudo–depth cue. Because Stage 2 selects removable regions by differencing the input and geometry-removed depth maps, this injected cue provides sufficient contrast for Stage 2 to identify and remove the watermark, showing that minimal conditioning tweaks let our method generalize to non-geometric inpainting cases.

# E CausRem

We construct the CausRem dataset by collecting 200 high-quality images from the free stock platform Pexels[3], including 100 images containing reflections and 100 with shadows. For each image, we manually annotate the primary object along with its causal visual effects—reflections or shadows.

In the shadow subset, where multiple objects often co-occur, we randomly select two to three objects per image for annotation. Each object and its corresponding shadow mask are stored using distinct IDs to preserve one-to-one causal relationship. In the reflection subset, due to the presence of fewer objects, we annotate only a single object-mask pair per image.

Fig. 11 and Fig. 12 illustrate representative annotation examples from the dataset.

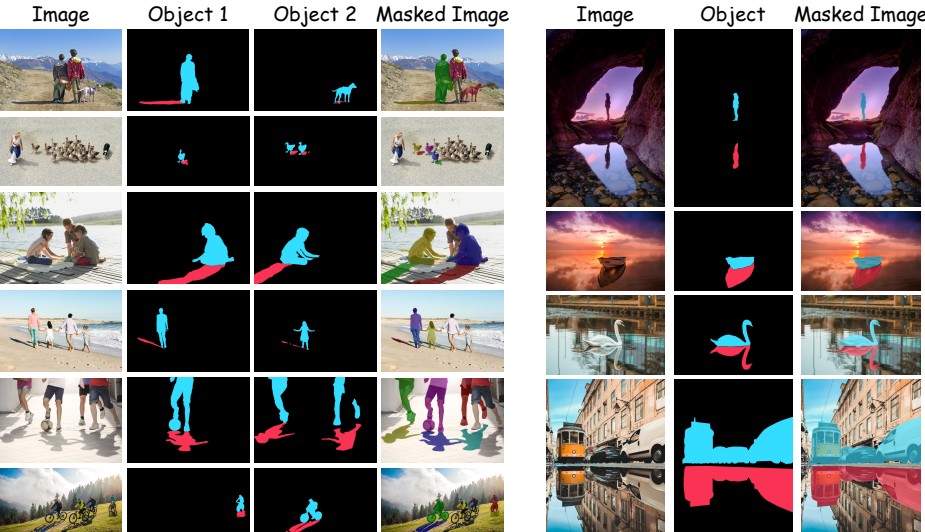

Figure 11: Representative annotations in CausRem. Left: shadow examples; Right: reflection examples.

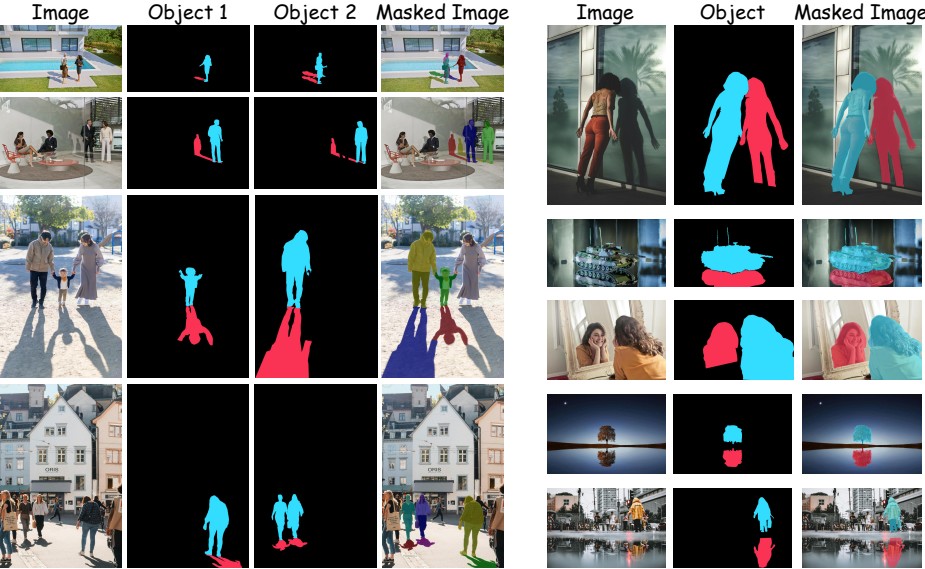

Figure 12: Representative annotations in CausRem. Left: shadow examples; Right: reflection examples.

---

[3]https://www.pexels.com

