# OpenReview forum: "GeoRemover: Removing Objects and Their Causal Visual Artifacts"
_NeurIPS.cc/2025/Conference — NeurIPS 2025 spotlight_

### Official Review · Reviewer_KFtu · 2025-06-16

**Clarity:** 4
**Significance:** 3
**Originality:** 3
**Rating:** 6
**Confidence:** 4

**Summary:**

This paper introduces a diffusion-based model for the removal of objects in images, in a way that also removes the artifacts that these objects create on the rest of the image, including reflections and shadows. The method relies upon a 2-step process: first, depth map object removal and then a cyclical diffusion model that performs both inpainting and removal simultaneously. The method is evaluated across several metrics, ablation studies, and comparisons with previous work, showing a significant advantage compared to baselines.

**Questions:**

- How does this model handle transparency and translucency?
- How does the model depend on the diffusion backbone used for the image generation?
- Is LORA a limiting factor in the model results?
- How expressive is the reflection removal? Does it handle color bleeding well?
- How does the model handle the removal of light-emitting objects (Eg  artificial light sources, bioluminescence, etc)?

**Ethical Concerns:**

["NO or VERY MINOR ethics concerns only"]

**Final Justification:**

I appreciate the rebuttal provided by the authors, which has successfully addressed most of my concerns. I highly suggest the authors to incorporate the perceptual metrics into the paper, which really add value to the analyses. I thus increase my rating to a 6 and therefore fully recommend acceptance.

The incorporation of perceptual metrics into the analysis, and the discussion about LORA, adds robustness to the paper.

**Limitations:**

A more comprehensive analysis of the model limitations would have benefited this paper.

**Paper Formatting Concerns:**

No formatting concerns.

**Quality:**

4

**Strengths And Weaknesses:**

Strenghts
- The paper tackles a very salient problem in computer vision, which is that of removing shadows and reflections on image inpainting and object removal algorithms.
- The proposed method is built upon sound assumptions on how geometry affects images, and how to consider them for adequate removal of objects.
- The method is sound, is well designed, and combines a smart use of existing components like LORA with particular advancements in the way geometry is completed with diffusion models using a reward-based framework. The second step of the method is very smart, and I think could inspire additional works on image-to-image translation and cycle-consistent generative models.
- The method is evaluated very well against previous models, with relevant metrics
- This paper is exceptionally well written, it is very easy to follow and the motivation is clear.
- Code and data will be shared upon publication, which should greatly improve reproducibility.
- Supplementary material contains a few additional examples on the ground truth labels.
- The results of the method are visually excellent.
- The paper introduces a benchmark dataset with 200 images for the object removal task, which, while limited in size, can be benefitial for the literature.

Weaknesses
- The paper tackles a somewhat niche problem, as it introduces a method to enhance the results of image inpainting algorithms in the cases where the objects to be inpainted are introducing additional artifacts in the image that are not necessarily present within the inpainting mask. These include shadows or reflections, which are important artifacts. However, I believe this feature also limits the scope of the method to some extent. By introducing this geometric prior, and the fact that the method heavily relies on depth maps, the method is limited to a set of objects and inpainting scenarios. Therefore, it may not be the optimal method for inpainting other types of images, including texture inpainting, watermarking removal, etc.
- The method relies upon pre-trained diffusion models, and therefore their quality may be dependent on their expressiveness.
- DreamSIM, SSIM, FLIP, DISTS and CLIP-IQA could have been additional useful metrics for the model evaluation, as they would provide additional insights on the model capabilities.
- On tables 1-3, for the appropriate metrics (eg LPIPS or PSNR), I think it would be useful to include confidence intervals. These require no additional training or any added computational cost.
- While I appreciate the use of the Aesthetics Score (AS) in the validation, I am not sure that it is measuring anything particularly related with object removal, as it does not seem to correlate with any other metric.
- On tables 1-3, I think it would be useful for clarity to show the second best results, as well as the worst results in red.
- I believe the L_flow should be ablated in more detail, as its impact is not particularly clear.
- Additional edge cases could have been interesting, including transparency (object removal through a piece of fabric or a window), reflections from light sources. etc.
- The method is computationally very expensive.

---

> ### Author Rebuttal · Authors · 2025-07-30
>
> ### The paper tackles a somewhat niche problem.
> **_Note: Dear Reviewer, we sincerely thank you for your time and effort in reviewing our work. Due to the official policy for this year’s rebuttal process, we are, unfortunately, unable to share the carefully prepared one-page document of additional visualizations. These include failure cases, intermediate results, and various edge case examples, which we found both interesting and thrilling to analyze, as they were inspired by the reviewers’ constructive comments. We also reached out to the AC to ask whether we could share this document in the same way as a code link, and the AC kindly consulted the program chairs on our behalf; however, we were informed that this is not allowed. Despite these constraints, we are making every effort to describe these visualizations in detail so that you can still obtain the corresponding information. These described figures are referred to using lettered indices (Case A–H). We will update our main submission with content from this document._**
>
> Our method indeed is not specifically designed to optimize texture inpainting or watermark removal. However, our model can also handle general inpainting tasks. In fact, it performs competitively on ordinary inpainting scenarios, while showing clear advantages in cases involving causal artifacts. To further demonstrate its applicability, we present a watermark removal example in Case H: a photo of a wooden dock where the watermark is uniformly distributed across both the lake surface and the dock planks. Our model successfully removed the watermark.
> ### The method relies upon pre-trained diffusion models & How does the model depend on the diffusion backbone used for the image generation?
> | Backbone | FID ↓ | CMMD ↓ | LPIPS ↓ | PSNR ↑ | AS ↑ | Insert. ↓ |
> |----------|--------|----------|-----------|-----------|--------|-------------|
> | SDXL     |    32.40    |     0.196     |    0.112       |    22.75       |   4.51     |      **1.35%**       |
> | FLUX.1-Fill (Ours) | **31.15** | **0.182** | **0.103** | **23.70** | **4.69** | 1.48% |
>
> We appreciate the reviewer’s insightful question. Like many diffusion-based approaches, our method inevitably depends on the expressiveness of the underlying pre-trained diffusion backbone, as the quality of Stage 2 image generation is influenced by its ability to render realistic textures and details. However, this is a common setting in current state-of-the-art methods such as CLIPAway, OmniEraser, OmniPaint, and ObjectDrop, all of which also rely on the same family of pre-trained diffusion models. Moreover, to provide a fair reference, we already explicitly reported the baseline performance of the pre-trained diffusion model (FLUX.1-Fill) itself, allowing readers to clearly understand how much improvement our two-stage framework brings beyond the base model’s capability.
>
> To quantify this dependency, we replaced the default Flux Fill backbone with SDXL Inpainting. As shown in the above table, Flux Fill only achieves slightly better performance on geometry-sensitive metrics, indicating that the overall performance of our method does not heavily rely on the specific diffusion backbone.
>
> ###  DreamSIM, SSIM, FLIP, DISTS and CLIP-IQA.
> | Method | SSIM ↑ | DreamSim ↓ | FLIP ↓ | DISTS ↓ | CLIP-IQA ↑ | SSIM ↑ | DreamSim ↓ | FLIP ↓ | DISTS ↓ | CLIP-IQA ↑ |
> |--------|--------|-------------|---------|-----------|--------------|--------|-------------|---------|-----------|--------------|
> |        | *RemovalBench* |        |         |           |              | *RORD-Val* |        |         |           |              |
> | CLIPAway | 0.6298 | 0.1572 | 0.1175 | 0.1656 | **0.4973** | 0.6074 | 0.1304 | 0.1645 | 0.1580 | **0.7986** |
> | Attentive-Eraser | 0.7084 | 0.0536 | **0.0854** | 0.1168 | 0.4790 | 0.7186 | 0.0878 | 0.1174 | 0.1243 | 0.7270 |
> | OmniEraser | 0.6367 | 0.0539 | 0.1084 | 0.1277 | 0.4339 | 0.6071 | 0.0675 | 0.1524 | 0.1325 | 0.6646 |
> | **Ours** | **0.7367** | **0.0304** | 0.0863 | **0.0770** | 0.4146 | **0.8248** | **0.0459** | **0.1026** | **0.0798** | 0.7807 |
>
> We appreciate the reviewer’s constructive suggestion. Following this advice, we added the suggested metrics for a more comprehensive evaluation. As shown in the above table, our method achieves the best performance on most of them across both RemovalBench and RORD-Val. Regarding CLIP-IQA, we note that, like the Aesthetics Score (AS), it primarily reflects overall image quality rather than object removal accuracy. While it may not strongly correlate with removal effectiveness, we include it for completeness.
>
> ###  On tables 1–3, for the appropriate metrics (e.g., LPIPS or PSNR), I think it would be useful to include confidence intervals.
> | Method | LPIPS ↓ (RemovalBench) | PSNR ↑ (RemovalBench) | LPIPS ↓ (RORD-Val) | PSNR ↑ (RORD-Val) |
> |--------|------------------------|------------------------|---------------------|-------------------|
> | CLIPAway | 0.254 ± 0.0337 | 18.78 ± 1.0025 | 0.278 ± 0.0134 | 16.36 ± 0.4525 |
> | Attentive-Eraser | 0.146 ± 0.0153 | 20.60 ± 1.0495 | 0.221 ± 0.0125 | 20.24 ± 0.5060 |
> | OmniEraser | 0.133 ± 0.0225 | 21.11 ± 0.9188 | 0.166 ± 0.0087 | 22.13 ± 0.4093 |
> | **Ours** | **0.124 ± 0.0209** | **25.52 ± 1.1277** | **0.103 ± 0.0067** | **23.70 ± 0.4612** |
> We have computed and reported the confidence intervals in the above Table.
>
> ###  Aesthetics Score.
>
> We will replace AS with more relevant metrics, such as those you suggested (DreamSIM, SSIM, FLIP, DISTS).
>
> ###  Show the second-best results, as well as the worst results in red.
>
> We appreciate the reviewer’s helpful suggestion. We will incorporate second-best and worst results in the revised version to improve clarity. However, due to the formatting restrictions of the OpenReview rebuttal system, we are unable to use colored fonts here.
>
> ###  More ablation study about L_flow.
>
> **_Note: due to space limitations, we may refer to responses addressed to other reviewers; we hope for your kind understanding._**
>
> We appreciate the reviewer’s constructive suggestion. More detailed ablation study is provided in the response to Reviewer w1kx, Questions **_DPO needs better motivation_** and  **_DPO lambda value_**.
>
>
> ### The method is computationally very expensive.
>
> We appreciate the reviewer’s comment on the computational cost. The detailed computation cost is provided in the response to Reviewer w1kx, Questions **_Computational Overhead & Inference clock time_**
>
> ### Edge cases (transparency and translucency)
>
> We appreciate the reviewer’s insightful question. The edge cases (Case B and Case C) for transparency and translucency can be found in our response to Reviewer Dwf1, Question **_How does the two-stage framework handle dynamic or translucent objects where depth maps are ambiguous?_**.
>
> ###  Is LoRA a limiting factor in the model results?
>
> We appreciate the reviewer’s question. LoRA is not a limiting factor in our framework. In fact, LoRA can already effectively adapt the backbone to our two-stage pipeline, which indicates that full fine-tuning would likely yield even better performance if sufficient computational resources were available. However, when the available training data is significantly smaller than the data used for pre-training—as is common in most downstream tasks—LoRA is often a better choice than full fine-tuning, as it reduces the risk of overfitting while achieving competitive or even superior generalization. Therefore, LoRA should not be considered a limitation in this context.
>
> ### How expressive is the reflection removal? Does it handle color bleeding well?
>
> We appreciate the reviewer’s question. Our method can handle most reflections without overlapping components but currently struggles with overlapping color bleeding. Specifically, for non-overlapping reflections, the model achieves strong performance, as also illustrated in our paper’s Figure 5.
>
> However, for complex reflections with overlapping components, such as color bleeding on highly reflective surfaces, our current model struggles. In Case C, a scene contains a large transparent water jug and five semi-transparent cups placed on a reflective table. The reflections of these cups overlap, causing noticeable color bleeding. When attempting to remove one of the semi-transparent cups, both the cup itself and its direct reflection was successfully removed. But the color bleeding in the reflections of the other cups, which was influenced by the removed cup, remains visible.
>
>
> ###  Edge cases (light-emitting objects)
>
> Our current model cannot effectively handle light-emitting objects. Case D presents a challenging self-emitting object removal case. Two light bulbs lie on the floor, one casting red light and the other green. Stage1 handles the geometry perfectly—both bulbs and their shapes are accurately removed in the depth map. However, Stage2 behaves unexpectedly: instead of simply erasing the red illumination, it “imagines” a diffuse, amorphous red glow appearing in midair, as if a new red light source had been added to the scene. Although the removal result is unsatisfactory, this behavior reveals an intriguing aspect of Stage 2’s reasoning. It has implicitly learned a causal relationship between illumination and its source—“believing” that if there is red light in the scene, there must be a corresponding red light source to explain it. This over-reasoning causes hallucinated artifacts in this case, yet it also suggests that our rendering model is not merely filling textures arbitrarily; instead, it captures some meaningful light–source correlations, which, if guided properly, could be advantageous in more physically consistent rendering tasks.
>
> ### Potential solution for all edge cases
> This limitation mainly arises from the lack of paired data for all edge cases, which we plan to address by synthesizing targeted training data using text-to-video generation models in future work, using prompts like _A hand removes a red light bulb from a table._

---

> > ### Comment · Reviewer_KFtu · 2025-08-04
> > **Response**
> >
> > I appreciate the rebuttal provided by the authors, which has successfully addressed most of my concerns. I highly suggest the authors to incorporate the perceptual metrics into the paper, which really add value to the analyses. I will increase my rating to a 6 and therefore fully recommend acceptance.

---

> > > ### Author Response · Authors · 2025-08-04
> > >
> > > We are especially grateful for the valuable suggestions regarding perceptual metrics, which have significantly strengthened our analysis. We will incorporate these metrics into the final version to enhance both the completeness and practical relevance of the work. We truly appreciate your support and are glad to hear that the revised version has addressed your concerns.

---

### Official Review · Reviewer_wwGK · 2025-07-01

**Clarity:** 2
**Significance:** 3
**Originality:** 3
**Rating:** 4
**Confidence:** 4

**Summary:**

This paper addresses the problem of object removal in 2D images. To better remove objects along with their shadows and mirrored reflections, the authors propose using depth as an intermediate representation in image inpainting. Specifically, they first inpaint the depth map and then reconstruct the image conditioned on the inpainted depth. This allows artifacts caused by reflections and occlusions to be removed at the depth level.
Specifically, the authors introduce a two-stage framework: The first stage performs depth inpainting. The second stage generates the final image conditioned on the inpainted depth. Both stages are trained separately based on FLUX and LoRA.

**Questions:**

I have two main concerns regarding this paper:
1. Handling of reflective objects: As shown in Figure 5, the method performs well on several mirror-like reflection cases. However, the paper does not include intermediate visualizations, which makes it somewhat difficult to understand which stage contributes most to the removal of reflection artifacts - whether it is the initial depth estimation, the depth inpainting stage, or the final image generation. Providing intermediate results from each stage could help me better understand the roles and effectiveness of each component in handling these challenging scenarios.
2. Lack of failure case analysis: While Tables 1 and 4 show that the proposed method outperforms all state-of-the-art baselines in terms of quantitative metrics, the paper would benefit from a more in-depth discussion of failure cases. In particular, it would be informative to include examples where the full model does not perform well, rather than only showing ablation results like those in Figure 4.
Understanding where and why the model fails would provide valuable insight into its limitations and potential areas for future improvement.

**Ethical Concerns:**

["NO or VERY MINOR ethics concerns only"]

**Final Justification:**

Thanks for the author's detailed response. Although the step-by-step visual results are not presented, I believe the author's analysis of the failure cases largely addresses my concerns. I will raise my score to borderline accept, and I hope the authors will include visualizations and further analysis in future versions.

**Limitations:**

The method relies on the Depth Anything model for depth estimation, which may limit performance in certain challenging scenarios, especially where depth prediction is inherently ambiguous.

**Paper Formatting Concerns:**

No concerns.

**Quality:**

3

**Strengths And Weaknesses:**

Strengths:
1. The idea of decoupling object removal into a two-stage process is novel and interesting. By leveraging two strong base models—Depth Anything and FLUX, the proposed approach achieves superior image generation quality, outperforming all state-of-the-art methods on standard benchmarks.
2. The paper is logically well-structured, and the training procedure for both stages is clearly explained, making it easy to follow.
3. The authors introduce a real-world test dataset, CausRem, which includes 200 challenging cases such as reflections and shadows.

Weaknesses:
The method relies heavily on the depth estimation quality of the Depth Anything model.

---

> ### Author Rebuttal · Authors · 2025-07-30
>
> ### The method relies heavily on the depth estimation quality of the Depth Anything model.
> We thank the reviewer for pointing out this important aspect. In our implementation, we use an off-the-shelf depth estimator solely to obtain a geometric representation.  Strictly speaking, our method does not rely on any specific depth estimator; rather, it assumes that an accurate geometric representation is available. Our core hypothesis and contribution lie in showing that, given a reasonably accurate geometry, object removal and its causal artifacts (e.g., shadows and reflections) can be effectively eliminated through our two-stage pipeline. Depth Anything is simply a strong depth estimator that helps us obtain more accurate geometry in practice, but our approach is not inherently tied to it. Moreover, the availability of many high-quality off-the-shelf depth estimators today makes it easy to obtain such geometry in practice.
>
> To further verify this, we replaced Depth Anything with a different depth estimator VGG-T (Wang, Jianyuan, et al. "Vggt: Visual geometry grounded transformer." Proceedings of the Computer Vision and Pattern Recognition Conference. 2025.).  As shown in the below Table, although the performance with VGG-T slightly drops compared to Depth Anything, our pipeline still achieves consistent artifact removal, confirming that our method can generalize to other depth estimators.
>
> | Depth Estimator | FID ↓ | CMMD ↓ | LPIPS ↓ | PSNR ↑ | AS ↑ | Insert. ↓ |
> |------------------|--------|---------|-----------|----------|--------|-------------|
> | VGG-T            | 31.27  | 0.185   | 0.107     | 23.68    | 4.35   | 1.71%       |
> | Depth Anything   | **31.15**  | **0.182**   | **0.103**     | **23.70**    | **4.69**   | **1.48%**       |
>
>
> ### Providing intermediate results from each stage for handling of reflective objects.
>
> We thank the reviewer for this valuable comment and apologize for not making this clearer in the paper. The intermediate reflections can actually be observed in Figure3(a) (rightmost boat example) in the paper. For instance, the boat’s reflection is estimated as having the same depth as the lake surface rather than a new boat-like geometry. Therefore, reflection artifacts are not present in the depth domain. After Stage1 removes the boat geometry, Stage2 leverages this updated geometry to simultaneously remove the boat and its corresponding reflection artifacts. In summary, the initial depth estimation and depth inpainting stage provide guidance on which objects need to be removed, while the final image generation stage removes the objects and their corresponding reflection artifacts.
>
> **_Note: Dear Reviewer, we sincerely thank you for your time and effort in reviewing our work. Due to the official policy for this year’s rebuttal process, we are, unfortunately, unable to share the carefully prepared one-page document of additional visualizations. These include failure cases, intermediate results, and various edge case examples, which we found both interesting and thrilling to analyze, as they were inspired by the reviewers’ constructive comments. We also reached out to the AC to ask whether we could share this document in the same way as a code link, and the AC kindly consulted the program chairs on our behalf; however, we were informed that this is not allowed. Despite these constraints, we are making every effort to describe these visualizations in detail so that you can still obtain the corresponding information. These described figures are referred to using lettered indices (Case A–H). We will update our main submission with content from this document._**
>
> We further demonstrate the robustness of our pipeline to depth estimation errors in Case G. The reflection of a duck on the lake surface is incorrectly estimated by the depth estimator as having a duck-like shape rather than the lake’s flat depth. Nevertheless, the final removal is still correct because the actual object depth (the duck itself) is accurate, enabling Stage2 to identify it as removable and also remove its reflection. This robustness arises because Stage 2 is trained on depth maps produced by Depth Anything, which naturally contain some depth estimation errors. According to our statistics, in approximately 85% of cases where reflection depths are estimated incorrectly, the final removal remains correct.
>
> ### Lack of failure case analysis.
>
> We appreciate the reviewer’s suggestion to elaborate on the failure cases. For Stage 2, we provide two representative failure cases in Case C and Case D.
>
> The Case C shows a semi-transparent case with a large water jug and five semi-transparent cups on a reflective table. The reflections of these cups overlap, causing noticeable color bleeding. When removing one of the cups, the model successfully removes both the cup itself and its direct reflection. However, the color bleeding in the reflections of the other cups, which was influenced by the removed cup, remains visible. This reveals a limitation of our current approach in handling reflective surfaces with complex, overlapping color interactions.
>
> The Case D presents a challenging self-emitting object removal case. Two light bulbs lie on the floor, one casting red light and the other green. Stage1 handles the geometry perfectly—both bulbs and their shapes are accurately removed in the depth map. However, Stage2 behaves unexpectedly: instead of simply erasing the red illumination, it “imagines” a diffuse, amorphous red glow appearing in midair, as if a new red light source had been added to the scene. Although the removal result is unsatisfactory, this behavior reveals an intriguing aspect of Stage 2’s reasoning. It has implicitly learned a causal relationship between illumination and its source—“believing” that if there is red light in the scene, there must be a corresponding red light source to explain it. This over-reasoning causes hallucinated artifacts in this case, yet it also suggests that our rendering model is not merely filling textures arbitrarily; instead, it captures some meaningful light–source correlations, which, if guided properly, could be advantageous in more physically consistent rendering tasks.
>
> To the best of our knowledge, no existing inpainting or object removal method can effectively handle such cases either, mainly due to the lack of paired data to provide explicit guidance. To address these limitations, we plan to leverage text-to-video generation models to synthesize paired data for such challenging edge cases, which we expect will improve the model’s robustness in future work.
>
> As for failure cases where Stage 1 struggles to remove the target geometry, they mainly occur when the user-provided mask is incomplete. In Case F, three semi-transparent glass bottles sit snugly inside a cardboard box. We attempt to remove the largest bottle on the left under two conditions: one with a complete mask fully covering the bottle, and another with an incomplete mask covering only about 70% of it. The difference is striking—when given the complete mask, Stage 1 neatly erases the bottle as expected. But with the partial mask, the model gets “confused” and hallucinates an extra bottle, as if trying to “fill in what it thinks should still be there.” This behavior, while undesirable for removal, actually reveals that the model has learned a non-trivial understanding of geometric structures: rather than blindly removing masked regions, it attempts to infer plausible object shapes based on surrounding geometry. Such behavior can be easily avoided in practice by applying simple mask dilation operations or using advanced segmentation models like SAM2 to provide complete masks.
>
> ### Limitations: The method relies on the Depth Anything model for depth estimation, which may limit performance in certain challenging scenarios, especially where depth prediction is inherently ambiguous.
> We thank the reviewer for pointing out this important aspect. As discussed above, our model is not restricted to any single depth estimator. Furthermore, recent progress in depth estimation has significantly improved robustness in challenging scenarios. For example, Choi et al. ("Self-supervised Monocular Depth Estimation Robust to Reflective Surface Leveraged by Triplet Mining," ICLR 2025) explicitly address reflective and complex surfaces. With such rapid advances, obtaining reliable geometry is becoming increasingly feasible, which we believe will further enhance the applicability and robustness of our proposed method.
>
> **_Note: due to space limitations, we may refer to responses addressed to other reviewers; we hope for your kind understanding._**
>
> Additionally, when depth estimation is imperfect, our pipeline includes mechanisms to compensate for such inaccuracies. A detailed example (Case A) is provided in our response to Reviewer Dwf1 Question **_How does the two-stage framework handle dynamic or translucent objects where depth maps are ambiguous?_**.

---

> ### Comment · Reviewer_wwGK · 2025-08-04
>
> As stated in my Final Justification, the analysis of the failure cases sufficiently addressed my concerns. Consequently, I have adjusted my score to Borderline Accept. Thank you for your detailed response.

---

> > ### Author Response · Authors · 2025-08-04
> >
> > Thank you very much for your thoughtful re-evaluation and for taking the time to carefully consider our rebuttal. We are glad to hear that the analysis of failure cases helped address your concerns. We will make sure to include the discussed failure cases and visual examples in the final version of the paper. We also apologize for not being able to share the visualizations during the rebuttal phase due to policy restrictions. Thank you again for your efforts and support throughout the review process.

---

### Official Review · Reviewer_w1kx · 2025-07-01

**Clarity:** 4
**Significance:** 4
**Originality:** 3
**Rating:** 5
**Confidence:** 5

**Summary:**

The paper presents a two-step diffusion pipeline for object removal that also erases the object’s shadows using depth as a geometry signal. In the 1st stage, a depth-aware diffusion model takes the masked image and its depth map, then learns to remove the object in the depth domain with DPO loss without artefacts. Next, a second diffusion model receives three inputs which are the original RGB image, the original depth map, and the object removed depth map. With these inputs, the second model produces the final image in which both the object and any shadow or reflections are removed.

**Questions:**

- How robust is both stages of the pipeline with respect to the possible incorrect depth maps? For example, in a failure case of depth anything in stage 1, how does the geometry removal of the object behaviour is effected for the model. Also, if the geometry removal of stage 1 is not perfect, how robust is the second stage pipeline to such changes? I think adding some slight noise to the original RGB images or increasing resolution which is explained as a failure case for the model [2] (In limitations section), and then executing the pipeline and analyzing how the scores have been changed can provide a better explanation in terms of the inner workings of the model.

- REMOVE [1] which is a metric that directly measures the object removal quality can also be used for analysis to better state the success of this paper compared to other baselines.

- I would like to see example failure cases for both stages of the pipeline. Currently the paper only includes success cases.

- What is the λ value used for the DPO coefficient in the stage 1 loss?

- What is the inference clock time overhead of this method compared to other baselines? This is stated as a limitation so can be better provided with numbers.

---

**I would like to see more failure case examples and robustness of the model as well as DPO's effectiveness compared to a more simple alternative. Given satisfactory answers, I would increase my score.**

---
[1] Chandrasekar, Aditya, et al. "Remove: A reference-free metric for object erasure." Proceedings of the IEEE/CVF Conference on Computer Vision and Pattern Recognition. 2024. [code](https://github.com/chandrasekaraditya/ReMOVE)
[2] Yang, Lihe, et al. "Depth anything: Unleashing the power of large-scale unlabeled data." Proceedings of the IEEE/CVF Conference on Computer Vision and Pattern Recognition. 2024.

**Ethical Concerns:**

["NO or VERY MINOR ethics concerns only"]

**Final Justification:**

I think this paper solves an open problem in a novel way. In rebuttal, I asked for failure case examples and robustness of the model as well as DPO's effectiveness compared to a more simple alternative. Given satisfactory answers (comparing dpo with tv loss and explaining some possible failure cases and robustness of the model), I had increase my score to an accept.

**Limitations:**

Limitations need more explanation. The authors state the high computational cost of 2 stage pipeline as a limitation (lines 509-514) which is correct but I think the failure cases as a limitation also needs to be discussed for both stages.

**Quality:**

3

**Strengths And Weaknesses:**

**Strengths**
- The paper is well written and easy to follow.
- The paper tries to solve a valid problem being present in the current literature. Significance is high.
- The paper made extensive comparisons with current literature including both GAN based and Diffusion based models as comparison.
- Using Depth as a supervision for removing shadows in supervision as well as two-stage decoupling with the DPO-guided depth loss is a novel approach to tackle the problem.

**Weaknesses**
- **Limitations need more explanation:**  The authors state the high computational cost of 2 stage pipeline as a limitation (lines 509-514) which is correct but I think the failure cases as a limitation also needs to be discussed for both stages.
- **Use of DPO despite a qualitative example in figure 3a needs a better explanation in terms of motivation:**
Figure 3a does illustrate that training **with** DPO yields cleaner geometry than training **without** it, yet this single visual comparison does not fully justify adopting a RL-style objective in place of simpler depth-smoothness regularizer such as total variation loss. Conceptual explanation of why DPO is more suitable for this task needs more expanding.
- **Computational overhead:** Due two 2 stage inference, this model is more computationally expensive compared to other baselines.

---

> ### Author Rebuttal · Authors · 2025-07-30
>
> ### Limitations need more explanation & Failure cases for each stage
> **_Note: Dear Reviewer, we sincerely thank you for your time and effort in reviewing our work. Due to the official policy for this year’s rebuttal process, we are, unfortunately, unable to share the carefully prepared one-page document of additional visualizations. These include failure cases, intermediate results, and various edge case examples, which we found both interesting and thrilling to analyze, as they were inspired by the reviewers’ constructive comments. We also reached out to the AC to ask whether we could share this document in the same way as a code link, and the AC kindly consulted the program chairs on our behalf; however, we were informed that this is not allowed. Despite these constraints, we are making every effort to describe these visualizations in detail so that you can still obtain the corresponding information. These described figures are referred to using lettered indices (Case A–H). We will update our main submission with content from this document._**
>
> We appreciate the reviewer’s suggestion to elaborate on the failure cases of both stages. For Stage 2, we provide two representative failure cases in Case C and Case D.
>
> The Case C shows a semi-transparent case with a large water jug and five semi-transparent cups on a reflective table. The reflections of these cups overlap, causing noticeable color bleeding. When removing one of the cups, the model successfully removes both the cup itself and its direct reflection. However, the color bleeding in the reflections of the other cups, which was influenced by the removed cup, remains visible. This reveals a limitation of our current approach in handling reflective surfaces with complex, overlapping color interactions.
>
> The Case D presents a challenging self-emitting object removal case. Two light bulbs lie on the floor, one casting red light and the other green. Stage1 handles the geometry perfectly—both bulbs and their shapes are accurately removed in the depth map. However, Stage2 behaves unexpectedly: instead of simply erasing the red illumination, it “imagines” a diffuse, amorphous red glow appearing in midair, as if a new red light source had been added to the scene. Although the removal result is unsatisfactory, this behavior reveals an intriguing aspect of Stage 2’s reasoning. It has implicitly learned a causal relationship between illumination and its source—“believing” that if there is red light in the scene, there must be a corresponding red light source to explain it. This over-reasoning causes hallucinated artifacts in this case, yet it also suggests that our rendering model is not merely filling textures arbitrarily; instead, it captures some meaningful light–source correlations, which, if guided properly, could be advantageous in more physically consistent rendering tasks.
>
> To the best of our knowledge, no existing inpainting or object removal method can effectively handle such cases either, mainly due to the lack of paired data to provide explicit guidance. To address these limitations, we plan to leverage text-to-video generation models to synthesize paired data for such challenging edge cases, which we expect will improve the model’s robustness in future work.
>
> As for failure cases where Stage 1 struggles to remove the target geometry, they mainly occur when the user-provided mask is incomplete. In Case F, three semi-transparent glass bottles sit snugly inside a cardboard box. We attempt to remove the largest bottle on the left under two conditions: one with a complete mask fully covering the bottle, and another with an incomplete mask covering only about 70% of it. The difference is striking—when given the complete mask, Stage 1 neatly erases the bottle as expected. But with the partial mask, the model gets “confused” and hallucinates an extra bottle, as if trying to “fill in what it thinks should still be there.” This behavior, while undesirable for removal, actually reveals that the model has learned a non-trivial understanding of geometric structures: rather than blindly removing masked regions, it attempts to infer plausible object shapes based on surrounding geometry. Such behavior can be easily avoided in practice by applying simple mask dilation operations or using advanced segmentation models like SAM2 to provide complete masks.
>
> ### DPO needs better motivation
>
> | Method | FID ↓ | CMMD ↓ | LPIPS ↓ | PSNR ↑ | AS ↑ | Insert. ↓ |
> |--------|--------|----------|----------|---------|------|------------|
> | TV     | 33.42  | 0.205    | 0.118    | 23.21   | 4.61 | 3.87%      |
> | DPO    | **31.15**  | **0.182**    | **0.103**    | **23.70**   | **4.69** | **1.48%**      |
>
> We appreciate the reviewer’s insightful question regarding the motivation for adopting a DPO-inspired objective. Total variation (TV) loss encourages local smoothness, which is effective for reducing depth noise but does not explicitly distinguish between correct geometry completion and implausible insertions. In contrast, DPO optimizes pairwise preferences between correct and hallucinated samples, directly aligning the model’s output with the desired global geometry. This is particularly important in ambiguous regions, where local smoothness may over-smooth fine structures or even propagate incorrect geometry, while DPO explicitly suppresses hallucinated insertions by learning from ranked preferences. Quantitatively, replacing DPO with a TV loss results in degraded performance, as shown in the above Table, confirming its effectiveness in maintaining geometric consistency.
>
> ### Computational Overhead & Inference clock time
>
> | **Method**        | **Time (s) ↓**
> |-------------------|----------------|
> | FLUX.1-Fill       | 7              |
> | CLIPAway          | 12             |
> | Attentive-Eraser  | 7              |
> | OmniEraser        | **6**          |
> | **Ours**          | 28             |
>
> We appreciate the reviewer’s comment on the computational cost. To further clarify this point, we provide a comparison of the inference clock time in Table above, which explicitly shows the runtime overhead of our two-stage pipeline relative to one-stage methods. In terms of runtime for editing a 512 × 512 image, our current implementation takes approximately 1 s for depth estimation, 7 s for geometry removal (Stage 1), and 20 s for appearance rendering (Stage 2) per image on a single H100 GPU. In practical deployment scenarios, various acceleration strategies—such as model distillation and architectural pruning—can be applied to significantly reduce inference time. Therefore, although the current runtime introduces some overhead, it does not pose a significant obstacle to practical applications. Moreover, although this cost is indeed higher than that of one-stage methods, Figure4 and Table2 in the paper provide experimental evidence supporting the necessity of the two-stage design, as it significantly improves both object removal quality and artifact suppression.
>
> ### How robust is both stages of the pipeline with respect to the possible incorrect depth maps?
> **_Note: due to space limitations, we may refer to responses addressed to other reviewers; we hope for your kind understanding._**
>
> We appreciate that the reviewer precisely identified the issue and suggested improvement direction. Our Stage2 relies on comparing the input and geometry-removed depth maps from Stage1 to locate the regions to be removed; thus, severe depth estimation errors in Stage1 can indeed limit the overall performance. Moreover, we adopt an inference-time strategy similar to adding depth noise to mitigate this issue. A detailed example (Case A) is provided in our response to Reviewer Dwf1 Question **_How does the two-stage framework handle dynamic or translucent objects where depth maps are ambiguous?_**.
>
> ### Increasing resolution
>
> | Resolution | FID ↓ | CMMD ↓ | LPIPS ↓ | PSNR ↑ | AS ↑ | Insert. ↓ |
> |------------|--------|----------|----------|---------|------|------------|
> | 512×512    | 32.92  | 0.191    | 0.116    | 22.85   | 4.12 | 2.37%      |
> | 1024×1024  | **31.15**  | **0.182**    | **0.103**    | **23.70**   | **4.69** | **1.48%**      |
> We thank the reviewer for this constructive suggestion. We have indeed trained and evaluated our model at a lower resolution of 512×512 in earlier experiments, and the results were clearly inferior to the current 1024×1024 version. It confirms that higher-resolution training substantially improves the model’s robustness, as the reviewer suggested.
>
> ### REMOVE metric
>
> | Method           | RemovalBench | RORD-Val |
> |------------------|---------------|------------|
> | **Ground Truth**     | **0.931**         | **0.8541**     |
> | CLIPAway         | 0.8489        | 0.8059     |
> | Attentive-Eraser | 0.9201        | 0.9164     |
> | OmniEraser       |       0.9236        | 0.9163     |
> | **Ours**         | **0.9302**    | **0.8578** |
> We appreciate the reviewer’s constructive suggestion. We conducted additional experiments using the REMOVE metric, and the results are reported in the above Table. Although the absolute REMOVE scores seem not perfectly calibrated (on RORD-Val, the Ground Truth only achieves 0.8541, which suggests that the metric may not be strictly upper-bounded by real images), our method achieves the score (0.8578) that is closest to the Ground Truth compared to all other baselines. This indicates that our two-stage framework produces object removal results that are perceptually more consistent with the real images.
>
> ### DPO lambda value
>
> We set λ = 0.1 in our final implementation. We also experimented with higher values (e.g., λ = 0.5 and λ = 1), but both led the model to converge to an “easy” solution, where the masked regions were overly smoothed and simply filled with uniform depth values, rather than preserving geometric details. The lower value of 0.1 provides a better balance between enforcing geometric consistency and maintaining fine structural details.

---

> ### Comment · Reviewer_w1kx · 2025-08-01
>
> Thank you for your responses, which have addressed my concerns. I will raise my score to a 5 and hope for acceptance. I would also like to thank the authors for their effort for trying to provide explanations with caption of images and tables as this year they are unable to provide visualizations.

---

> > ### Author Response · Authors · 2025-08-03
> >
> > We are grateful to Reviewer w1kx for the constructive feedback and for raising the score. We appreciate your understanding of the visualization policy this year, and we are glad that our efforts to enhance the captions and textual explanations were helpful in addressing your concerns. We will revise our paper accordingly based on your valuable suggestions. Thank you again for your thoughtful review and support of our work.

---

### Official Review · Reviewer_Dwf1 · 2025-07-13

**Clarity:** 4
**Significance:** 4
**Originality:** 4
**Rating:** 5
**Confidence:** 3

**Summary:**

The paper introduces GeoRemover, a two-stage framework for object removal that explicitly addresses the challenge of eliminating both target objects and their causal visual artifacts e.g., shadows, reflections. The first stage removes objects from geometric representations, depth maps, using strictly mask-aligned training, ensuring structural accuracy. The second stage renders a photorealistic RGB image conditioned on the updated geometry, depth map, implicitly removing artifacts. A preference-driven objective, inspired by DPO, is introduced to prevent hallucinations during geometry removal. Experiments on benchmark datasets shows state-of-the-art performance in both object and artifact removal.

The paper presents a significant advancement in object removal by addressing causal artifacts through a principled, two-stage approach. While computational costs and paired data requirements are limitations, these are common trade-offs in state-of-the-art methods and do not detract from the paper’s contributions.

**Questions:**

1. How does the two-stage framework handle dynamic or translucent objects where depth maps are ambiguous?
2. What are the trade-offs between the strict mask alignment in Stage 1 and the flexibility of loose alignment in prior work?

**Ethical Concerns:**

["NO or VERY MINOR ethics concerns only"]

**Final Justification:**

I appreciate the authors' detailed thoughtful response. All my concerns got addressed fairly. I keep the final score "Accept".

**Limitations:**

1. High Resource Usage: The two-stage design increases computational overhead, compared with one-stage methods.
2. Depth Estimator Dependency: Performance is tied to the quality of the depth map, which may introduce errors in low-texture or occluded regions.
3. Artifact Scope: The method focuses on shadows and reflections; how about other artifacts, such as motion blur

**Quality:**

4

**Strengths And Weaknesses:**

Strengths:
1. Novel Two-Stage Framework: Decoupling geometry removal and appearance rendering enables precise structural editing and implicit artifact removal, addressing a critical challenge occurred in previous work.
2. Preference-Driven Training: The DPO-inspired loss effectively suppresses hallucinations and ensures geometric consistency, improving controllability compared to loosely mask-aligned methods.
3. SOTA performance: GeoRemover outperforms existing methods on multiple metrics (FID, PSNR, IoU) across two benchmarks, including a new CausRem dataset for causal artifacts.

Weaknesses:
1. Computational Cost: The two-stage pipeline increases runtime and memory usage compared to one-stage methods.
2. Reliance on Depth Estimation: The method depends on external depth estimators, whose accuracy could degrade in complex scenes.
3. Paired Data Requirement: The rendering stage requires paired data (images with and without objects/artifacts), which may not always be available in real-world scenarios.

---

> ### Author Rebuttal · Authors · 2025-07-30
>
> ### Computational cost & High resource usage.
>
> We appreciate the reviewer’s comment on the computational cost. In terms of runtime, our current implementation takes approximately 1 s for depth estimation, 7 s for geometry removal (Stage 1), and 20 s for appearance rendering (Stage 2) per image (512 * 512) on a single H100 GPU. We would like to emphasize that our implementation has not yet been optimized for latency. In practical deployment scenarios, various acceleration strategies—such as model distillation and architectural pruning—can be applied to significantly reduce inference time. Therefore, although the current runtime introduces some overhead, it does not pose a significant obstacle to practical applications.
> Regarding memory usage, the overall peak memory consumption is 34GB, which poses no practical limitation for deployment on current GPU hardware.
>
> ### Reliance on depth estimation & Depth estimator dependency.
>
> We appreciate the reviewer’s insightful comment regarding the reliance on depth estimation. Our method indeed assumes accurate geometry as a prerequisite, and performance may degrade if the depth estimation fails in complex scenes. Nevertheless, recent progress in this field has significantly improved robustness in such scenarios. For example, Choi et al ("Self-supervised Monocular Depth Estimation Robust to Reflective Surface Leveraged by Triplet Mining." ICLR. 2025.) explicitly address reflective and complex surfaces. With such rapid advances, obtaining reliable geometry is becoming increasingly feasible, which we believe will further strengthen the applicability of our method.
>
> ### Paired data requirement.
>
> We appreciate the reviewer’s valuable comment on the paired data requirement. While collecting perfectly paired data can be challenging in some real-world scenarios, recent works have shown that such data can be efficiently extracted from videos (e.g., RORD and OmniEraser), and most state-of-the-art object removal methods also rely on paired supervision. Moreover, with the rapid progress in text-to-video generation, creating paired data is becoming increasingly practical. For example, a simple prompt such as “A red apple sits on a table; a hand enters the frame and takes the apple away” can generate frames with and without the object, naturally forming paired training samples.
>
> ### How does the two-stage framework handle dynamic or translucent objects where depth maps are ambiguous?
> **_Note: Dear Reviewer, we sincerely thank you for your time and effort in reviewing our work. Due to the official policy for this year’s rebuttal process, we are, unfortunately, unable to share the carefully prepared one-page document of additional visualizations. These include failure cases, intermediate results, and various edge case examples, which we found both interesting and thrilling to analyze, as they were inspired by the reviewers’ constructive comments. We also reached out to the AC to ask whether we could share this document in the same way as a code link, and the AC kindly consulted the program chairs on our behalf; however, we were informed that this is not allowed. Despite these constraints, we are making every effort to describe these visualizations in detail so that you can still obtain the corresponding information. These described figures are referred to using lettered indices (Case A–H). We will update our main submission with content from this document._**
>
> We appreciate the reviewer’s insightful question. Our Stage 2 model determines which regions to remove by comparing the difference between the original and the geometry-removed depth maps. For dynamic or translucent objects, depth estimation can be incomplete, resulting in little or no difference between the two depth maps and thus limiting Stage 2’s effectiveness.
>
> In Case A, three individuals are running, and motion blur caused by high-speed movement leads to erroneous depth estimation by Depth Anything. Specifically, the depth of one runner’s foot is completely misestimated as part of the background. Consequently, during Stage 1 geometry removal, this foot region lacks valid depth information, and the Stage 1 output becomes almost identical to the input depth map.
>
> Since Stage 2 relies on comparing the input and output depth maps from Stage 1 to identify the regions requiring removal, no removal occurs when both maps are nearly identical. In this case, the absence of reliable geometric guidance prevents Stage 2 from performing effective object removal.
>
> Fortunately, this issue can be easily addressed by filling the missing depth values within the masked region using the maximum depth value from its local neighborhood (e.g., a 10×10 pixel window). Surprisingly, simply propagating the maximum depth value from nearby pixels—“borrowing depth from neighbors”—almost magically restores the missing foot geometry, enabling Stage 2 to confidently erase the leg. In addition, missing depth values can be detected by checking whether the depth within the masked region remains almost unchanged before and after geometry removal.
>
> After applying this Local Max Depth Fill-in strategy, the previously missing foot depth is filled with the maximum depth values from its local neighborhood, producing a pseudo-complete depth map and significantly enhancing the depth contrast between the masked region and its surroundings. Stage 2 can thus accurately identify the foot region as an object to be removed, resulting in the correct elimination of the runner’s leg and a visually consistent reconstruction of the ground surface. This demonstrates that even a simple geometric completion strategy can substantially improve the robustness of our two-stage framework under challenging dynamic conditions.
>
> We also provide transparent and semi-transparent examples in Case B and Case C. The Case B shows a fully transparent glass sphere placed on grass. The distorted, magnified grass patch is particularly interesting, as it appears as though the grass “jumps forward” in depth, confusing the estimator. However, as long as the depth of the grass exists, our model can still remove the magnified grass behind the glass sphere. The Case C shows a large transparent water jug and five semi-transparent cups on a reflective table. What is fascinating here is that despite the complex light interactions on the reflective table, the model precisely identifies and removes the cups with its reflection. These results demonstrate that our model is reasonably robust to both transparent and semi-transparent objects. We will include these findings and discussions in the final version of the paper.
>
> ### What are the trade-offs between the strict mask alignment in Stage 1 and the flexibility of loose alignment in prior work?
>
> There is an inherent trade-off between strict and loose mask alignment. During training, strict mask alignment provides clear geometric guidance, explicitly indicating which regions should be modified and which should remain unchanged. However, its limitation becomes evident during inference: if the user provides an imperfect mask—e.g., failing to fully cover a cup—the uncovered parts will not be removed. In contrast, loose mask training improves robustness to low-quality masks by allowing the model to modify regions outside the given mask, reducing its reliance on precise mask inputs. The downside, as illustrated in Figure 1 of the paper, is that it often alters unintended regions, causing uncontrolled geometry modifications.
>
> We therefore do not adopt loose mask training, as our strict alignment framework combined with mask-level augmentation can already reduce the dependence on high-quality masks during inference while avoiding unintended geometry modifications. Instead, following SmartEraser, we introduce mask-level data augmentation under the strict alignment framework, including mask dilation, boundary perturbation, and slight morphological expansion. This augmentation exposes the model to imperfect masks during training, enabling it to tolerate minor mask inaccuracies while retaining the structural precision guaranteed by strict mask alignment.  In practice, we observe that as long as the mask covers approximately 95% of the target object, our method can still achieve reliable geometry removal, demonstrating robustness to small inaccuracies in mask coverage.
>
> ### Artifact scope.
>
> We appreciate the reviewer’s question. Motion blur is indeed a causal artifact induced by object motion and thus, in principle, fits within the scope of our geometry-driven framework. However, our current training lacks paired data that explicitly provides motion blur supervision, which limits its performance in such scenarios. We observed in Case A, after the runner’s leg is removed, the model does not attempt to erase the motion blur associated with the leg. Instead, it interprets the occluded region as if it originally contained motion blur and “faithfully completes” that blurred streak, treating it as part of the static background rather than as an artifact to be removed. This limitation is common to all existing object removal methods, as paired supervision for motion blur removal is rarely available. As more paired datasets or video-based supervision become available, our two-stage design should naturally generalize to handle motion blur due to its reliance on geometric priors rather than artifact-specific heuristics. We are very interested in exploring this further as our next step.

---

> > ### Comment · Area_Chair_iDPL · 2025-08-05
> >
> > Dear reviewer Dwf1,
> >
> > The authors have provided extensive discussion on the model tradeoffs, hard cases like dynamic or translucent objects and more. As the Author-Reviewer discussion is coming to an end, we appreciate your participation in the discussion. Did the the authors answer your concerns? Does it change your score?
> >
> > -- Your AC

---

### Note · Authors · 2025-08-13

We would like to sincerely thank the AC for facilitating communications with the program chairs and for carefully following up on our request to share additional visualizations. Although this year’s policy did not allow us to provide the prepared figures, your support and guidance helped us adapt our rebuttal to convey all requested information in detailed textual form. We are also grateful to all reviewers for their constructive feedback, which led us to enrich the paper with comprehensive failure-case analyses, intermediate-stage discussions, additional perceptual metrics, confidence intervals, and deeper explanations of our design choices.

In our rebuttal, we addressed key points raised by the reviewers, including: (1) robustness to dynamic, transparent, and semi-transparent objects, introducing a Local Max Depth Fill-in strategy and analyzing challenging light-emitting cases; (2) handling of reflection artifacts and color bleeding, with limitations identified for complex overlapping reflections; (3) the trade-off between strict and loose mask alignment, and how strict alignment combined with mask augmentation improves robustness without sacrificing precision; (4) the motivation and quantitative advantage of the DPO-inspired objective over simpler smoothness losses; (5) generalization to different depth estimators; (6) computational cost analysis with acceleration strategies; and (7) extended experiments with additional perceptual metrics and confidence intervals. We will incorporate these additions into the final version of the paper to ensure the main text fully reflects these clarifications and improvements.

This process has made our work more complete and better presented. At last, we truly appreciate the AC’s efforts and the thoughtful engagement from all reviewers, which have significantly improved the quality and clarity of our submission.

---

### Decision · Program_Chairs · 2025-09-17

**Decision:**

Accept (spotlight)

**Comment:**

The paper proposes a geometry-aware method for object removal from images. First, the approach proposes to remove the object from the depth using a preference-driven objective. As a second step, the method renders the image conditioned on the updated depth . This work addresses a long-standing issue of high-quality object removal in images, as it is notoriously hard to identify and remove causal effects like shadows or reflections from the image.

The authors acknowledge that the limitation of the method is that it is hard to handle transparent or dynamic objects, Light-emitting objects or other cases where depth is ambiguous. Throughout the rebuttal, the authors added a new removal metric, perceptual metrics, inference time estimates, ablations on preference-based training, mask-level augmentation.

The reviewers agree on the strong contributions of the paper.